# Response outcomes gate the impact of expectations on perceptual decisions

Ainhoa Hermoso-Mendizabal [1,7], Alexandre Hyafil [1,2,6,7], Pavel E. Rueda-Orozco [3], Santiago Jaramillo [4], David Robbe [5] & Jaime de la Rocha [1✉]

Perceptual decisions are based on sensory information but can also be influenced by expectations built from recent experiences. Can the impact of expectations be flexibly modulated based on the outcome of previous decisions? Here, rats perform an auditory task where the probability to repeat the previous stimulus category is varied in trial-blocks. All rats capitalize on these sequence correlations by exploiting a transition bias: a tendency to repeat or alternate their previous response using an internal estimate of the sequence repeating probability. Surprisingly, this bias is null after error trials. The internal estimate however is not reset and it becomes effective again after the next correct response. This behavior is captured by a generative model, whereby a reward-driven modulatory signal gates the impact of the latent model of the environment on the current decision. These results demonstrate that, based on previous outcomes, rats flexibly modulate how expectations influence their decisions.

[1] Institut d'Investigacions Biomèdiques August Pi i Sunyer (IDIBAPS), Barcelona 08036, Spain. [2] Center for Brain and Cognition, Universitat Pompeu Fabra, Ramón Trias Fargas, 25, 08018 Barcelona, Spain. [3] Instituto de Neurobiología, UNAM, 76230 Santiago de Querétaro México, Mexico. [4] Institute of Neuroscience, University of Oregon, 1254 University of Oregon, Eugene, OR 97403, USA. [5] Aix Marseille Univ, INSERM, INMED, 63 Avenue de Luminy, 13009 Marseille, France. [6] Present address: Centre de Recerca Matemàtica, Campus de Bellaterra, 08193 Bellaterra, Spain. [7] These authors contributed equally: Ainhoa Hermoso-Mendizabal, Alexandre Hyafil. ✉email: jrochav@clinic.cat

Imagine Rafa Nadal returning Roger Federer's serve in the decisive game of a Grand Slam final. Serving at 185 km per hour, Nadal has a few hundred milliseconds to visually estimate the ball trajectory, prepare the motor plan, including where he aims to return the ball and execute it. In such speeded decisions based on partial or ambiguous sensory information, the anticipation provided by an informed prior expectation can be decisive because subjects can respond faster. Based on past games bringing the two players together, and on the pattern of the last serves executed by Federer, Nadal inevitably forms an expectation about where the next ball will arrive. Combined with the visual motion of the ball, this expectation may allow him to gain some decisive tens of milliseconds in the return of the serve[1]. However, if his prediction fails and he concedes an ace, does he need to choose between trashing his prior model on Federer's serve or sticking to it in the subsequent point? Or can Nadal transiently downplay the weight of his prediction on the next serve without modifying his prior?

Normative theories describe how prior expectations and ambiguous stimulus evidence should be combined in order to maximize categorization performance[2,3]. In dynamical environments, in which the statistics of the sensory information varies with time, subjects must be constantly updating their internal model by accumulating past stimuli, actions, and outcomes[4]. The updating of the prior based on the actions occurring in each trial typically introduces sequential effects, which are systematic history-dependent choice biases reflecting the impact of the trial-to-trial variations in expectation[5–16]. However, there are circumstances where subjects seem able to quickly and flexibly modulate the impact of prior expectations in driving their choices. One of such examples is the switch between (1) exploiting choices which, according to their current statistical model of the environment, are more likely to yield reward and (2) exploring alternative choices that are not aimed to maximize reward given that internal model, but to reduce environmental uncertainty and eventually refine the current model[17–19]. In particular, when the task design potentiates stochastic exploration, rats are able to operate in an expectation-free mode, in which choices did not depend on previous history[20]. In other tasks, the updating of the internal prior is not done in a continuous manner as new information is presented, but subjects update their internal estimates abruptly and intermittently when they feel there has been a change point in the environment[21]. Recent studies have shown that, in the absence of feedback, the magnitude of the expectation bias on current choice is smaller after a low confidence response[7,22,23]. Despite these findings, we lack a conceptual framework that could explain both how expectations are formed and which are the factors that regulate their use on a moment to moment basis.

Here we investigate whether the combination of expectation and sensory evidence can be dynamically modulated. Moreover, we aim to develop a unified model that jointly describes the dynamics of expectation buildup and its modulatory variables on a trial-by-trial basis. We train rats to perform perceptual discrimination tasks using stimulus sequences with serial correlations. Behavioral analysis allows us to tease apart the different types of history biases. In particular, rats accumulate evidence over previous choice transitions, defined as repetitions or alternations of two consecutive choices, in order to predict the next rewarded response. Crucially, this expectation-based bias disappears after an error, reflecting a fast switch into an expectation-free categorization mode. This switch does not imply, however, the reset of the accumulated expectation, which resumes its influence on behavior as soon as the animal obtains a new reward. This ubiquitous behavior across animals is readily captured by a nonlinear dynamical model, in which previous outcomes acts as a gate for the impact of past transitions on future choices.

## Results

**A reaction-time auditory task promoting serial biases.** To study how the recent history of stimuli, responses, and outcomes influence perceptual choices, we trained rats in a two-alternative forced choice (2AFC) auditory discrimination task, in which serial correlations were introduced in stimulus trial sequences (Fig. 1a–c)[7,24–26]. This design mimicked the temporal regularities of ecological environments and allowed us to probe the trial-by-trial expectations that animals formed about upcoming stimuli based on the changing statistics of the stimulus sequence. The serial correlations between trials were created using a two-state Markov chain (Fig. 1b) parameterized by the probability to repeat the previous stimulus category $P_{rep}$ (the unconditioned probabilities for each of the two categories were equal). We varied $P_{rep}$ between repeating blocks, in which $P_{rep} = 0.7$, and alternating blocks, in which $P_{rep} = 0.2$ (Fig. 1c; block length 200 trials). By poking into the center port, rats triggered the presentation of the stimulus, which lasted until they freely withdrew from the port. Each acoustic stimulus was a superposition of a high-frequency and a low-frequency amplitude-modulated tones, and animals were rewarded for correctly discriminating the tone with the higher average amplitude. The discrimination difficulty of each stimulus, termed stimulus strength $s$, was randomly and independently determined in each trial, and set the relative amplitude of each tone (Fig. 1b, d). When stimulus strength $s$ was null, i.e., contained no net evidence in favor of either alternative, the rewarded side was still determined by the outcome of the random Markov chain generating the stimulus category sequence (Fig. 1b).

**Across-trial dynamics of history-dependent choice biases.** Animals in Group 1 ($n = 10$ animals) completed an average of 508 trials per session (range 284–772 average trials), gathering an average of 56,242 trials in total per animal (range 15,911–81,654 trials). Psychometric curves, showing the proportion of Rightward responses as a function of the stimulus evidence, did not depend on block type (Fig. 2a, left). The horizontal shift of this psychometric curve, parametrized by the fixed side bias $B$, measured the history-independent preference of the subject toward one side. To estimate the impact on choice of the serial correlations of the stimulus sequence, we also analyzed the repeating psychometric curve (Fig. 2b). This new psychometric curve showed the proportion of trials where the animals repeated the previous choice as a function of the repeating stimulus evidence, i.e., the evidence favoring to repeat (when positive) or alternate (negative) the previous choice (see "Methods" for details). The horizontal shift of this curve, parametrized by the repeating bias $b$, measured the history-dependent tendency to repeat or alternate the previous choice. All animals developed a block-dependent repeating bias $b$ (Fig. 2a right, b left): after-correct trials, $b$ was positive in the repetitive block, and negative in the alternating block. Interestingly, the fixed side bias $B$ was similar across blocks for each animal (Fig. 2c, left), showing that animals' side preference was independent of the changes in repeating bias caused by block switching. Surprisingly, in trials following an error, $b$ almost vanished in both block types (Fig. 2b, c, right). Thus, after errors rats did not use previous history to guide their decision (mixed-effects ANOVA with factors *block* (repeating/alternating), congruent sequence length $n$, *previous outcome* (correct/error) and random effect *animal index* gave a significant interaction *block* × $n$ × *previous outcome* $F_{(6,250)} = 3.06$, $p = 0.007$; separate analysis yielded a significant interaction *block* × $n$ for the after-correct condition ($F_{(6,117)} = 33.66$, $p < 1e\text{-}22$), but not for the after-error condition ($F_{(6,117)} = 1.14$, $p = 0.34$)). In contrast, the fixed side bias $B$ was analogous after correct and error trials (Supplementary Fig. 1b).

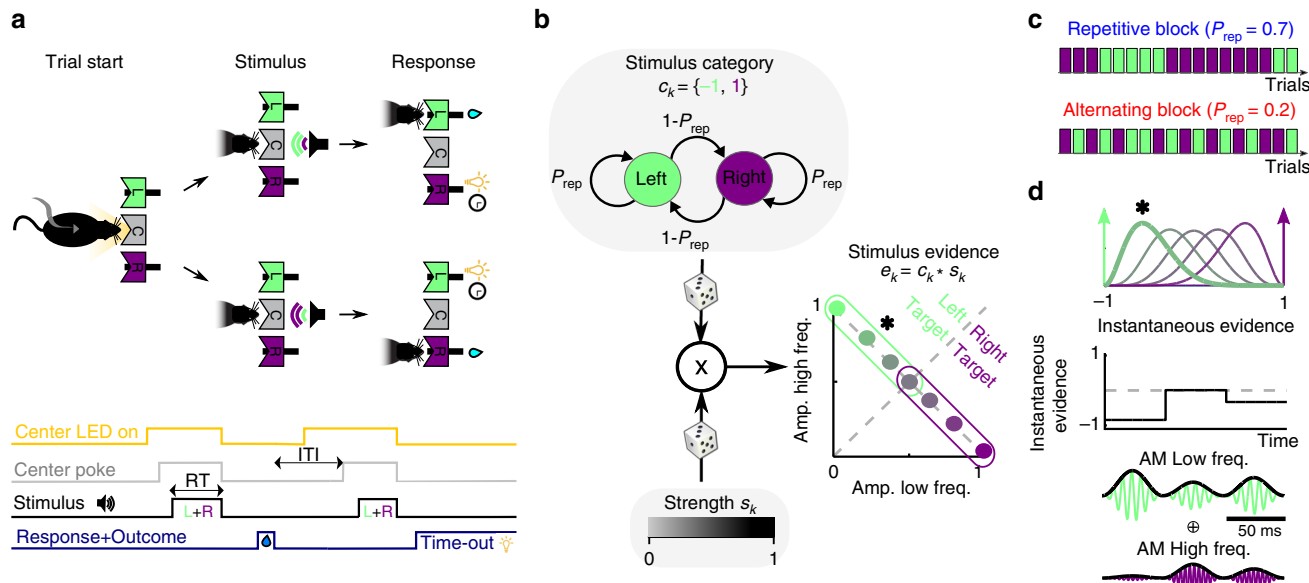

**Fig. 1 Auditory discrimination task and stimulus sequence statistics. a** Sketch of one trial of the task: cued by center port LED, rats poke in the center port to trigger the presentation of a mixture of two AM tones, each of which is associated with reward in the left (L) or right (R) port. Correct responses are rewarded with water, and incorrect responses are punished with a light plus a 5-s timeout. RT, reaction time. **b, c** Serial correlations in the sequence of stimuli were introduced by setting the probability of repeating the previous stimulus category $P_{rep}$ (top in **b**) in blocks of 200 trials named repetitive block and alternating block (**c**). The stimulus strength $s_k$ was randomly drawn in the kth trial (bottom in **b**) to yield the stimulus evidence $e_k$, that determined the distance to the categorization boundary, i.e., the discrimination difficulty of the stimulus (right in **b**). **d** The stimulus evidence $e_k$ determined the distribution (top) from which the instantaneous evidence was drawn in each frame of the sound envelope (see color match with **b**). An instantaneous evidence trace (middle) and the AM modulated tones that result (bottom) are shown for an example stimulus with $e = -0.48$ (asterisks in **b** and **d**).

The expectation did not only affect rats' choices but also modulated their reaction times (Supplementary Fig. 1c, d). After correct trials, the reaction time was shorter for expected stimuli (i.e., trials in which the repeating stimulus evidence was congruent with the block's tendency) compared with unexpected stimuli (i.e., trials in which the repeating stimus evidence was incongruent with the block's tendency; ANOVA *block × repeating stimulus category* $F(1,126) = 134.59$, $p < 1e-6$; mean normalized RT(expected)-RT(unexpected) $= 0.10$, post hoc two-tailed paired $t$ test $p < 1e-10$). Crucially, after error trials the reaction time was not modulated by expectation (ANOVA, *block × repeating stimulus category × previous outcome* $F(1,264) = 26.77$, $p < 1e-6$; separate analysis for the after-error condition yielded *block × repeating stimulus category* $F(1,126) = 0.02$, $p = 0.88$)). Hence, as for choices, the impact of repeating bias $b$ on reaction time depended on previous trial outcome, being absent after error trials.

Rats used history information by tracking several trials over short windows into the past: the magnitude of the repeating bias $b$ built up with the number of consecutive correct past repetitions or alternations $n$ until it plateaued after $n = 5-10$ trials (Fig. 2e, blue and red line). Importantly, however, irrespective of $n$, the repeating bias $b$ reset almost completely with a single incorrect response for all rats (Fig. 2e black lines). The reset occurred independently of the strength of the incorrectly categorized stimulus (Supplementary Fig. 2a), and it only occurred after errors but not after correct but unexpected responses, e.g., one alternation after several repetitions (Supplementary Fig. 3a). To control that, the reset was not caused by forgetting due to the 5 s timeout imposed after errors, we trained a subset of rats using shorter random time-out durations (range 1–5 s) and found that the bias reset was maintained independently of time-out duration (Supplementary Fig. 2b, right). We also sought for other dependencies of $b$ on the length of the intertrial interval (ITI).

After correct choices, $b$ increased to more positive values for longer ITIs (Supplementary Fig. 2b, left), but the sudden decrease to near-zero values after errors occurred for all ITIs (Supplementary Fig. 2b, c; see below). Together, these observations suggest that rats updated their beliefs about the environment on a trial-by-trial basis and that this update crucially relied on the outcome of the preceding trials: longer sequences of rewarded repetitions/alternations led to stronger response prior, but one error was sufficient to make the animals abandon this prior independently of the time elapsed from previous rewards.

The trial-to-trial updating of the response prior had a direct impact on the animals categorization accuracy. Overall, the repeating bias seemed advantageous for the task, as the average categorization accuracy was higher for trials following a correct trial than for trials following an error, in which $b$ was reset to zero (0.76 versus 0.72, $p < 1e-04$ two-tailed paired $t$ test). However, the repeating bias increased the subjects' accuracy when it was congruent with the block tendency, but it decreased accuracy when it was incongruent with it (Supplementary Fig. 4a–d). Moreover, in the congruent condition, the impact of the prior on accuracy was largest for low stimulus strength and it vanished to zero as the stimulus strength increased. Thus, when stimuli were hard to categorize, expectations benefitted animals the most (see Supplementary Fig. 4 for details).

**GLM analysis on the integration of recent history information.** Having identified that rats used previous responses and outcomes to guide decisions, we aimed to identify the specific factors in trial history generating this repeating choice bias. In particular, these factors could be (1) a lateral bias that creates an attraction or repulsion toward the left or right side, depending on previous responses (Fig. 3a) and (2) a transition bias that creates an attraction toward repeating or alternating, depending on the

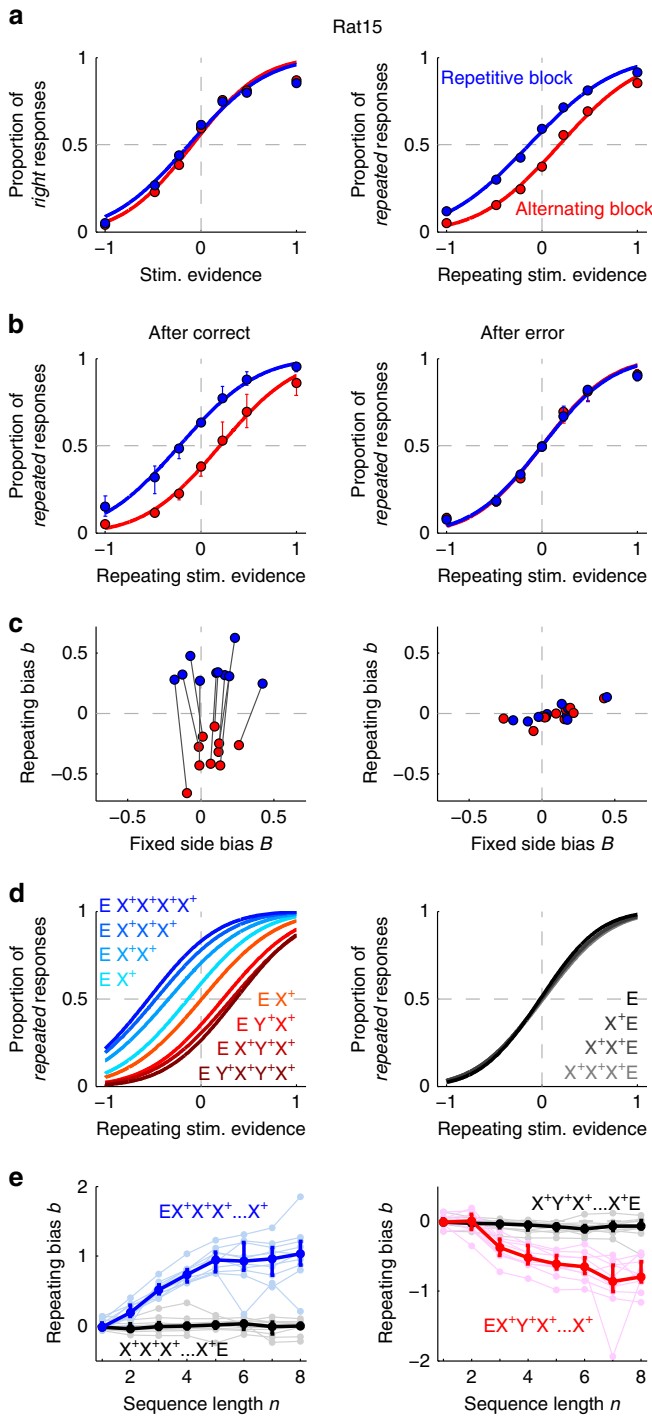

**Fig. 2 Build-up and reset dynamics of the repeating bias. a** Psychometric curves for an example animal showing the proportion of rightward responses vs. stimulus evidence (left) or of repeated responses vs. repeating stimulus evidence (right) computed in the repetitive (blue dots) or alternating blocks (red dots; color code applies for all panels). This animal shows a block-independent rightward fixed side bias $B > 0$ (left), and a block-dependent repeating bias $b$ matching the tendency of each block (right). Curves show fits using a probit function. **b** Proportion of repeated responses (median across $n = 10$ animals) computed in trials following a correct (left) or an incorrect response (right). **c** Repeating bias $b$ versus fixed side bias $B$ in the two blocks after a correct (left) or an incorrect response (right). Each pair of connected dots represents one animal. **d** Left: Fits of the proportion of repeated responses following trial sequences made of a different number of correct repetitions (blue gradient) or alternations (red gradient; see insets for color code). $X^+$ and $Y^+$ represent either rightward or leftward correct responses. E represents an error. Time in the sequences progresses from left to right. Right: same curves obtained when the sequence of correct repetitions is terminated by an error. **e** Repeating bias versus the length of the sequence of correct repetitions (left, blue) or alternations (right, red). Sequences terminated by an error are shown in black. Dark traces show median across animals while light traces show individual animals. Error bars show SD (**a**) or first and third quartiles (**b**, **e**).

transitions sequence is first accumulated into the transition evidence $z^T$, an internal estimate of the probability of the next transition, which in this example points the subject to predict a repetition in the last trial (Fig. 3c). Importantly, the transition evidence $z^T$ needs to be converted into an effective decision bias by projecting it into the right–left choice space (Fig. 3c, d). This is achieved in our framework by multiplying $z^T$ with the last response $r_{t-1}$, yielding the transition bias $\gamma^T = z^T \times r_{t-1}$ (see gray arrow in Fig. 3c, d). In this example, lateral and transition biases have an opposite influence in the final choice: while $\gamma^L$ has a rightward influence, $\gamma^T$ has a leftward influence because the transition evidence $z^T$ predicts a repetition and the last choice was leftward (compare Fig. 3a, d). Thus, the two biases extract different information from the sequence of past trials. Although only the transition bias is adaptive in the task, since such bias allows to take advantage of the serial correlations in the stimulus sequence in both types of blocks, the two biases could in principle contribute to the repeating bias $b$ described above.

To quantitatively assess how subjects computed these biases, we used an explorative approach that assumed that, in each trial, animals combined linearly the responses $r$ ($r = R, L$) and transitions $T$ ($T = Rep, Alt$) from the last ten trials in order to generate the lateral and transition bias, respectively (see gray boxes in Fig. 3b and Supplementary Fig. 5; see Supplementary Methods section 2 for details). The two biases were then combined with the stimulus evidence in order to yield a decision (Fig. 3e). Fitting such a generalized linear model (GLM) to the behavior of a rat implied finding the weight with which each of these past (e.g., previous transitions) and current events influenced the animal choices[5,7,10,13,15,22]. Because correct and error choices presumably had a different impact (Fig. 2b–d, right), we separated the contribution to the lateral bias $\gamma^L$ of rewarded responses $r^+$, sometimes called reinforcers[27,28], from error responses $r^-$ (Supplementary Fig. 5). Following the same rationale, we separated the contributions to the transition bias $\gamma^T$ of two consecutive correct responses ($T^{++}$) from transitions where either the first ($T^{-+}$), the second ($T^{+-}$), or both responses ($T^{--}$) were incorrect. After fitting the regression weights of the GLM individually for each rat, a consistent pattern across animals emerged (Fig. 4 orange curves; Supplementary Fig. 6a–d). The contribution to $\gamma^L$ of each response depended on whether the

history of previous repetitions and alternations (Fig. 3c). To understand the difference between these first-order (lateral) and second-order (transition) biases, we first considered correct responses only, and described the effect of errors below. If subjects were using e.g., the last four choices to estimate the probability of the next stimulus category, given the example choice sequence $R^+R^+R^+L^+$, where $R^+$ and $L^+$ represent a rightward or leftward correct choice ($L^+$ represents the last trial, Fig. 3b), they would estimate that $R$ is more likely and develop a lateral rightward bias $\gamma^L$ in the next trial (Fig. 3a). The same four-choice sequence can however be represented as the series of transitions $Rep^{++}Rep^{++}Alt^{++}$, where $Rep^{++}$ and $Alt^{++}$ represent repetitions and alternations between two correct responses. These

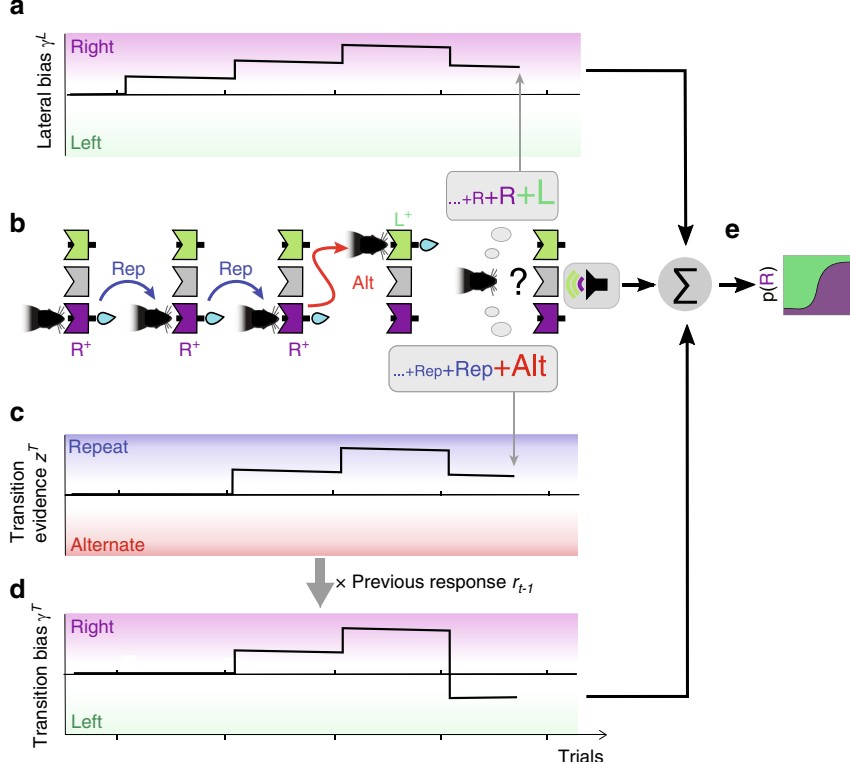

**Fig. 3 Dissecting two different history choice biases.** Cartoon of an example series of four choices, $R^+R^+R^+L^+$, illustrating the buildup of the lateral and transition biases. **a** The lateral bias, capturing the tendency to make rightward or leftward responses, increases toward the right in the first three $R^+$ trials, and compensates this buildup with the last $L^+$ response. Its net impact on the final trial is a rightward bias. **b** Schematic of the sequence of rewarded responses showing the transitions, defined as the relation between two consecutive responses, being repetitions (Rep, blue arrows) or alternations (Alt, red arrow). The animal computes each choice from combining its expectation based on a weighted sum of previous alternations (bottom gray balloon) and previous responses (upper gray balloon) with the current stimulus sensory information (see last trial). **c** Transition evidence $z^T$ captures the tendency to repeat or alternate the previous response based on the series of previous transitions $Rep^{++}Rep^{++}Alt^{++}$ predicting a repetition in the final trial. **d** The transition bias $\gamma^T$ is obtained by projecting the transition evidence $z^T$ (**c**) onto the right–left choice space via a multiplication with the previous response $r_{t-1}$ (see gray arrow). **e** The evidence provided by the current stimulus is summed to the addition of the biases $\gamma^L_t + \gamma^T_t$ and passed through a sigmoid function, yielding the probability of selecting a rightward response (Supplementary Fig. 5).

response was rewarded or not, following a win-stay-lose-switch pattern: while rats displayed a tendency to opt again for the side of previously rewarded responses (positive $r^+$ weights), they tended to opt away from previously non-rewarded responses (negative $r^-$ weights; Fig. 4a, orange curves). Similarly, previous transitions between two correct responses $T^{++}$ were positively weighted (Fig. 4b, c, orange curves), meaning that recent $++$ repetitions increased the tendency to repeat (positive impact on $\gamma^T$), while recent $++$ alternations increased the tendency to alternate (negative impact on $\gamma^T$). However, the transitions $T^{+-}$, $T^{-+}$, and $T^{--}$ with at least one error barely influenced subsequent choices (Fig. 4b). This means that, in the example choice sequence $R^+R^+R^+R^-$, equivalent to the transition sequence $Rep^{++}Rep^{++}Rep^{+-}$, only the first two repetitions impacted on $\gamma^T$. Thus, the only effective transitions driving the transition bias were $++$ transitions.

Error responses had yet a more dramatic effect on the transitions bias. They not only made the $T^{+-}$, $T^{-+}$, and $T^{--}$ transitions ineffective but they also suppressed the impact of all previously accumulated $T^{++}$ transitions: the weights of previous $T^{++}$ transitions were completely vanished when we fitted the GLM only using choices following an error trial (Fig. 4b, c, black curves). Thus, after an error choice, the transition bias was reset to zero, $\gamma^T = 0$, meaning that rats' behavior was completely blind to the history of previous repetitions and alternations, and was driven only by sensory information and lateral bias. The reset of

$\gamma^T$ was not an idiosyncratic strategy followed by some of our animals, but it was found in every animal we trained (Fig. 4c; Supplementary Fig. 7). In fact the magnitude of $T^{++}$ kernel was much more homogenous across animals than the lateral kernel (two-tailed $F$ test after-correct $T^{++}$ vs. $r^+$, $F(9,9) = 0.24$, $p < 0.05$; after-correct $T^{++}$ vs. $r^-$ $F(9,9) = 0.094$, $p < 0.002$; see Supplementary Fig. 7). The reset effect was, however, not observed in the lateral bias, which was moderately affected by errors (Fig. 4a, black curves). Furthermore, only the lateral kernels showed a dependence on the ITI (Supplementary Fig. 2d). Thus, the bias reset following errors was specific to the transition term, robust across different ITIs and extremely reliable across subjects.

Despite the strong impact of the transition bias, animal choices mostly relied on the current stimulus, which had an impact an order of magnitude larger than the transition bias, which was itself an order of magnitude larger than lateral bias (Supplementary Fig. 6e). The weakest (yet very consistent) sequential component was a stimulus repulsive bias reminiscent of an after-effect caused by sensory adaptation with a very slow recovery (Supplementary Fig. 6b). A modified analysis separating the effects of repetitions and alternations showed that they had largely symmetrical effects, suggesting that animals summarized both types of transition into a single rule that could take positive or negative values (Supplementary Fig. 8c). Importantly, the weights were identical when computed separately in repetition and alternation blocks (Supplementary Fig. 9) or for the trials at

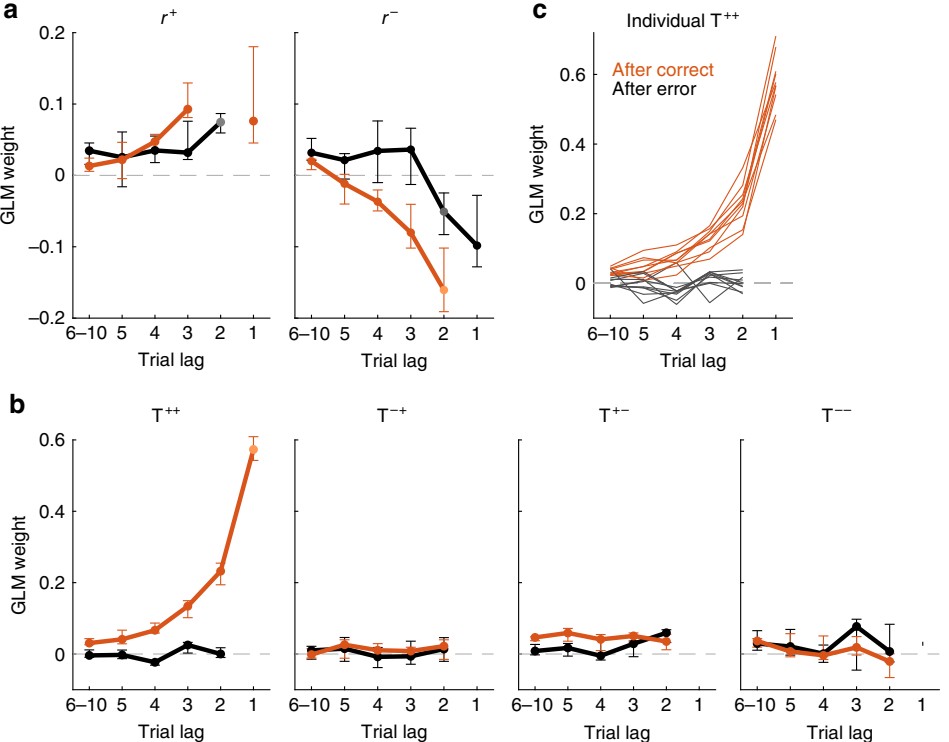

**Fig. 4 Fitted weights quantifying the impact of the lateral and transition biases onto animals decisions.** Influence of past events on current choice when separately fitting the choices in trials after a correct (orange) or an error response (black). **a** GLM weights of previously rewarded ($r^+$, left panel) and unrewarded ($r^-$, right panel) responses. These kernels quantify the influence on choice of the side (left vs. right) of previous responses. **b** Weights of previous transitions (repetition vs. alternation) computed separately for $T^{++}$ (a rewarded trial followed by a rewarded trial), $T^{-+}$ (error-rewarded), $T^{+-}$ (rewarded-error), and $T^{--}$ (error-error). Strong positive weighting of $T^{++}$ transitions after-correct responses revealed that animals tended to reproduce previous repetitions and alternations between two consecutive rewarded trials. Points in **a** and **b** show median coefficients across animals ($n = 10$) and error bars indicate first and third quartiles. **c** Transition kernels for individual animals show the ubiquity across subjects of the reset of the kernel after errors. Some weights at trial lags 1 and 2 are not shown because of existing indeterminacies between regressors (see Supplementary Methods Section 2.3).

the beginning of the blocks when the accuracy was smaller (average accuracy of trial ranges 1–50 and 50–200 was 0.73 and 0.76, two-tailed paired $t$ test $p < 1e-5$; Supplementary Fig. 10). This suggests that rats adopted a single strategy across all blocks, and the different repeating choice bias found in each block (Fig. 2b–e) simply reflected the difference in the statistics of the stimulus sequence (Fig. 1c). Because the impact of transitions decayed in around five trials (Fig. 4b left), the strategy allowed animals to switch the sign of their repeating bias relatively fast when switching between blocks (Supplementary Fig. 1a) at the cost of suffering relatively large fluctuations in the repeating bias within each block. Model comparisons further confirmed that the full model fitted separately for trials following correct trials and errors provided a better fit to rats' decisions than the full model fitted to all trials, or alternative models where the lateral and/or transition module were removed (Supplementary Fig. 8a). Importantly, the GLM with only lateral biases yielded a non-monotonic kernel for the lateral responses, a result that could lead to spurious interpretations when the effect of previous transitions was not considered (Supplementary Fig. 8b).

To test the extent to which these findings depended on the task design, we trained a new group of rats (Group 2, $n = 6$) in a different level discrimination 2AFC task, in which noise stimuli had to be classified according to the intensity difference between the two lateral speakers[29]. The stimulus sequence followed the same pattern as before with repeating and alternating blocks (Fig. 1b, c). Performing the same GLM analysis in this task yielded qualitatively the same results, including the reset of the transition bias after errors (Supplementary Fig. 11). Finally, we

found that the presence of a transition bias and its reset after errors was not contingent on the presence of serial correlations in the stimulus sequence. A third group of rats (Group 3, $n = 9$) exposed to only an uncorrelated stimulus sequence, exhibited the same qualitative pattern for the impact of previous transitions, although of smaller magnitude (Supplementary Fig. 12c). Once the sessions started featuring serial correlations in the stimulus sequence (Fig. 1b, c), the magnitude of the transition weights increased (Supplementary Fig. 12c), suggesting that the transition bias is an intrinsic behavior of the animals, but its magnitude can be adapted to the statistics of the environment. In total, these analyses show that the dependence on previous outcome of history-dependent biases is a general result across animals and across different tasks.

**Transition evidence is blocked, but not reset after an error.** We then asked whether the reset of the transition bias after errors reflected (i) a reset of the transition accumulated evidence $z^T$, meaning the entire system monitoring transitions underwent a *complete reset* (Fig. 5a); or whether, in contrast, (ii) information about previous transitions was maintained in $z^T$, but was gated off from causing a transition bias (Fig. 5b). Whereas in the latter scenario (gating hypothesis), the information maintained in $z^T$ could be used to compute $\gamma^T$ following new correct responses, in the complete reset scenario the buildup of both $z^T$ and $\gamma^T$ started back from zero following errors. To test these two hypotheses, we defined the transfer coefficient $\gamma^T_t \rightarrow \gamma^T_{t+k}$ that quantified how well we could predict the bias $\gamma_{t+k}^T$ in trial $t + k$ from the value of

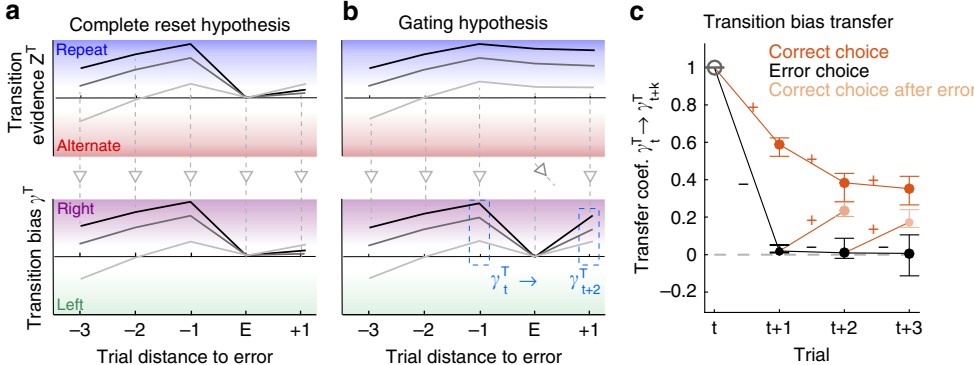

**Fig. 5 Transition bias is reset after errors but accumulated transition evidence is maintained. a, b** Schematics showing three example traces of the transition accumulated evidence $z^T_t$ (top) and transition bias on current response $\gamma^T_t$ (bottom) in two hypothetical scenarios. To facilitate visualization, the pattern of responses in all example traces was R+R+R+ER+. **a** Complete reset hypothesis: after an error at $t$, both variables reset $z^T_{t+1} \simeq \gamma^T_{t+1} \simeq 0$. Evidence $z^T_{t+2}$ is then built up de novo, implying that biases before ($\gamma^T_t$) and after ($\gamma^T_{t+2}$) the reset are independent. **b** Gating hypothesis: after an error, evidence $z^T_{t+1}$ is maintained but it does not convert into a bias, leading to the reset $\gamma^T_{t+1} \simeq 0$. After a correct response at $t+1$, the conversion is recovered and the value $\gamma^T_{t+2}$ correlates with $\gamma^T_t$. In the example shown, this implies that the sorting of $\gamma^T_t$ across traces is maintained for $\gamma^T_{t+2}$. **c** Transfer coefficient $\gamma^T_t \to \gamma^T_{t+k}$ versus trial lag $k$ quantifies the degree to which the transition bias at trial $t$ is predictive of the bias on subsequent trials (blue dashed boxes in **b**). It is calculated separately depending on the outcome of each trial (colored lines show rewarded choices and black lines error choices; see the section 2.6. in Supplementary Methods for details). While the transfer coefficient vanishes after errors (i.e., reset of the bias; black dots), a correct response following an error (light orange) brings it close to the value obtained when there are no errors (dark orange dots). This implies that the information about the value of the bias $\gamma^T_t$ is maintained when the bias is reset (i.e., gating hypothesis).

the bias $\gamma^T_t$ in trial $t$ (see section 2.6 in Supplementary Methods for details). The transfer coefficient was computed as a function of trial lag $k$ conditioning on the sequence of outcomes (Fig. 5c). When trial $t$ was correct, the bias was passed on to $t + 1$ with a discounting decay that mirrored the shape of the transition kernel in the GLM analysis (Fig. 5c, dark orange dots). The same discounting occurred going from $t + 1$ to $t + 2$ when trial $t + 1$ was correct. By contrast, if $t$ was incorrect, because of the bias reset after errors, the value of $\gamma^T_t$ was not predictive of the decision at trial $t + 1$, nor at trial $t + 2$ if $t + 1$ was also incorrect (Fig. 5c, black dots). Crucially though, the bias $\gamma^T_t$ in trial $t$ strongly influenced choices at trial $t + 2$ if trial $t$ was incorrect but trial $t + 1$ was correct (Fig. 5c, light orange dots). Its impact was significantly larger than zero for all rats (Wald test $p < 0.003$ for each of the $n = 10$; Supplementary Fig. 6g) and close in magnitude to the impact when both trials $t$ and $t + 1$ were correct. This rebound in choice predictability was even observed at $t + 3$ after two incorrect responses followed by a correct one (Wald test $p < 0.05$ for nine out of the $n = 10$). These results are consistent with the gating hypothesis (Fig. 5b), in which errors do not cause a reset of the accumulated transition evidence $z^T$ but do cause a transient cut off in the influence of $z^T$ on choice, visible as a reset in $\gamma^T$. This influence became effective again once the animal made a new correct response giving rise to the measured correlation between the values of the bias before and after the reset (Fig. 5b, gray vertical arrows; Supplementary Fig. 6h). An equivalent analysis on the lateral bias $\gamma^L$ showed that the bias transferred to the subsequent trials with a rapid decay, which was moderately affected by the outcome of the trials and showed no evidence of reset-and-rebound dynamics (Supplementary Fig. 6f).

**A dynamical model of history-dependent outcome-gated biases.** Having found that the transition bias underwent reset-and-rebound dynamics, we built a generative model that could implement the gating hypothesis. One latent variable in the model was the accumulated transition evidence $z^T$, which was updated in each trial, depending on whether the last choice was a repetition or an alternation and therefore maintained a running estimate of the transition statistics[7,8,10,13] (Fig. 6a). The

dependence of the leak of $z^T$ on the previous outcome could in principle implement the Complete reset hypothesis (Fig. 5a), if the leak following errors was complete ($\lambda_T \simeq 1$). A second modulatory variable $c^T$ modulated the influence of the transition evidence onto the current decision by setting the transition bias equal to $\gamma^T = c^T \times z^T \times r_{t-1}$. Importantly, $c^T$ was updated after each trial, based on the trial outcome. In addition to the transition bias, the model also featured accumulated lateral evidence $z^L$ that directly resulted in a lateral bias (i.e., $\gamma^L = z^L$).

We fitted the model parameters to the series of choices made by each rat (Fig. 6b–g; Supplementary Fig. 13) and obtained results in agreement with the gating hypothesis: first, correct transitions ($++$) led to strong changes in the transition evidence $z^T$, while the other transitions ($+-, -+, --$) did not lead to any consistent pattern (Fig. 6d). Second, the update parameters for $c^T$ corresponded to a vanishing of this variable after errors for at least seven rats out of ten, and a very strong recovery after any correct trial (Fig. 6f). This effectively converted the variable $c^T$ into a gating variable that was able to completely block the use of the accumulated transition evidence $z^T$ after a single error (Fig. 6g). By contrast, the leak of $z^T$ was not significantly different after correct trials and after errors ($p > 0.6$, paired $t$ test, two-tailed), providing further evidence that the reset of the transition bias did not correspond to a loss of the accumulated evidence, as predicted by the Complete reset hypothesis (Fig. 5a top). Third, correct rightward (leftward) responses increased the lateral bias in favor of the rightward (leftward) response (Fig. 6b). Fourth, model comparison showed that this dynamical model gave a better account than versions where either $c^T$ or the lateral bias $z^L$ were omitted, as well as of the GLM described in the previous section (Supplementary Fig. 14). Finally, adding a modulatory variable $c^L$ to the lateral module only had a marginal impact on model performance (Supplementary Fig. 15).

**Generative model simulation versus experimental data.** Finally, we assessed the capacity of the compact dynamical model to account for the dynamics of the previously reported repeating bias $b$ (Fig. 2d, e) by comparing model simulations to actual rat data. The model very closely reproduced the build-up dynamics of $b$ in

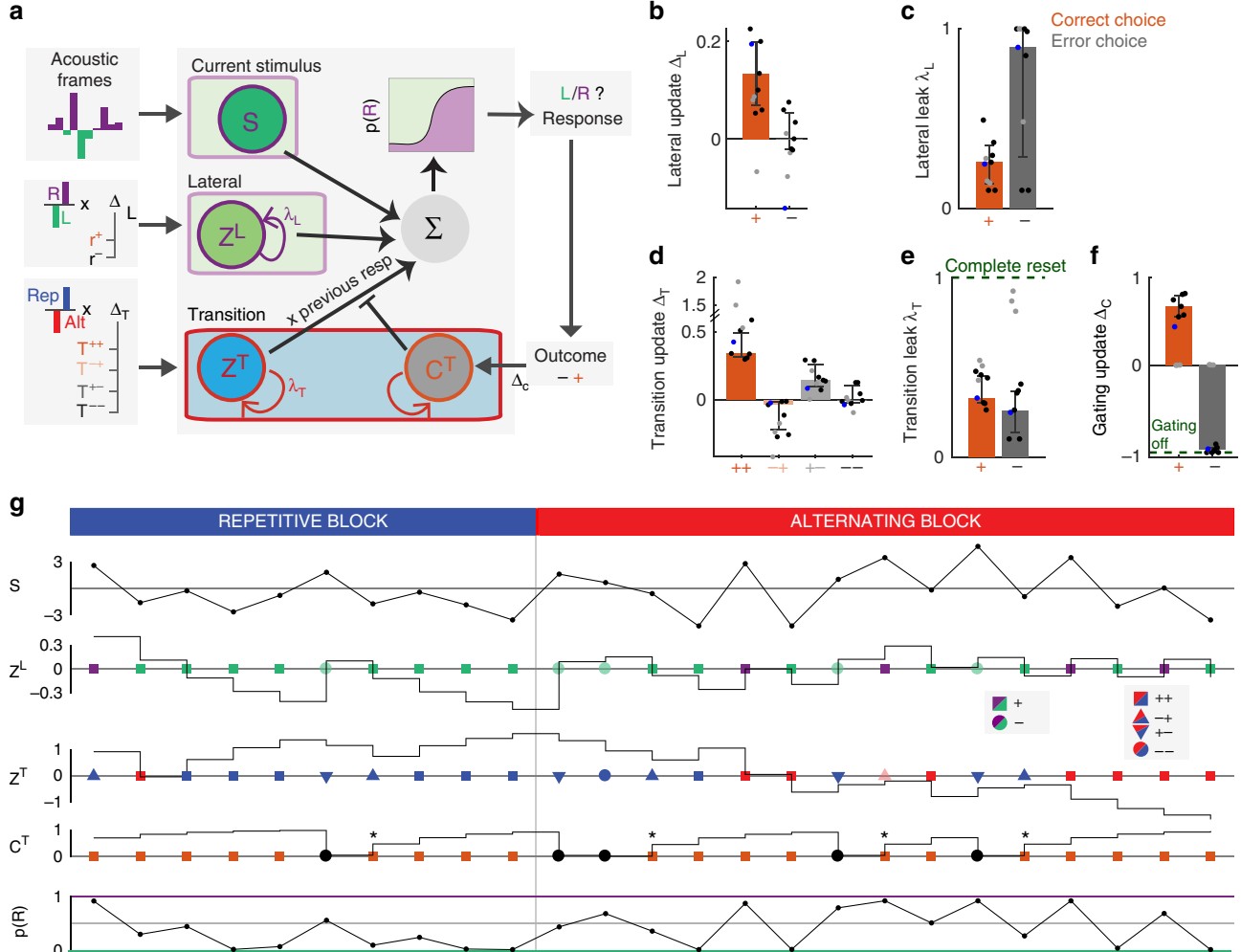

**Fig. 6 Dynamic generative model of history-dependent perceptual decisions. a** Architecture of the model. The sensory module accumulates the instantaneous stimulus evidence of the current trial. The lateral module maintains the lateral evidence $z^L$, which is updated depending on the last response and its outcome (updates $\Delta_L$). The transition module maintains the transition evidence $z^T$, which is updated depending on the last transition (i.e., $++,-+,+-,--$; updates $\Delta_T$). The modulatory signal $c^T$ is updated based on the last outcome (updates $\Delta_C$). The transition bias is obtained from the product $\gamma^T = z^T \times c^T \times r_{t-1}$. The sum of sensory evidence and the lateral and transition biases determined the probability to choose either response at the current trial. Parameters were fitted to the choices of each rat separately. **b–e** Best-fitting values of the model parameters. Bars show median across seven rats (black and blue points). Three rats were excluded from the statistics because the fitted model yielded a solution without gating dynamics (gray points). **b** Lateral evidence outcome-dependent update $\Delta_L$. **c** Outcome-dependent leak of the lateral bias $\lambda_L$. **d** Transition evidence update $\Delta_T$, depending on the outcome of the last two trials ($++, -+, +-, --$). **e** Outcome-dependent leak of the transition bias $\lambda_T$. **f** Outcome-dependent update of the transition gating signal $\Delta_C$. A value of $-1$ corresponds to an extinction of the gating signal on the subsequent trial (i.e., a full blockade of the corresponding bias), while $+1$ corresponds to full recovery of the bias (i.e., gating equal to its maximum value of 1). **g** Example traces of the dynamics of the latent variables across 25 trials switching from a repetition to an alternation block (fitted parameters correspond to rat 12 shown as blue points in **b–e**). Traces depict the variables stimulus evidence $S$, $z^L$, $z^T$, $c^T$, and overall probability to choose a rightward response. Symbols on the corresponding trial axis represent inputs to the variables $z^L$, $z^T$, and $c^T$: left (green) vs. right (purple) responses; repeating (blue) vs. alternating (red) transitions; and rewarded (orange) vs. error (black) outcomes. Symbols shape represent different outcome combinations (see inset). Notice the reset of $c^T$ after errors and the maintenance of $z^T$ afterwards (asterisks).

series of correct repetitions and alternations (Fig. 7a). Moreover, the model allowed to partition the value of $b$ into the contributions of the lateral and transition biases. While the transition bias was perfectly symmetric in series of repetitions and alternations (blue curves in Fig. 7a), the lateral bias behaved very differently: it only built up during series of repetitions, in which all the responses were on the same side, while it oscillated around zero in series of alternations, in which the contribution of each response was partially canceled by the next one (green curves in Fig. 7a). Thus the dissection of the repeating bias into the lateral and transition contributions explained the overall asymmetry found between the two blocks. In particular, the block asymmetry in $b$ after the first

correct choice (i.e., after an EX$^+$ sequence) could be explained by the two contributions having the same or opposite sign (see $n = 2$ in Fig. 7a). A similar argument applied for the asymmetries in $b$ found after correct, but unexpected responses (Supplementary Fig. 3b). Model simulations also reproduced the reset of repeating bias when a series of correct repetitions/alternations was interrupted by an error (Fig. 7b), and the subsequent rebound when the rat performed correctly again (Fig. 7c). Impressively, the model replicated the asymmetry in the magnitude of this rebound between the repeating and alternating blocks by summing (Fig. 7c, top) or subtracting (Fig. 7c, bottom), respectively, contributions of transition and lateral biases. Furthermore, the model provided a

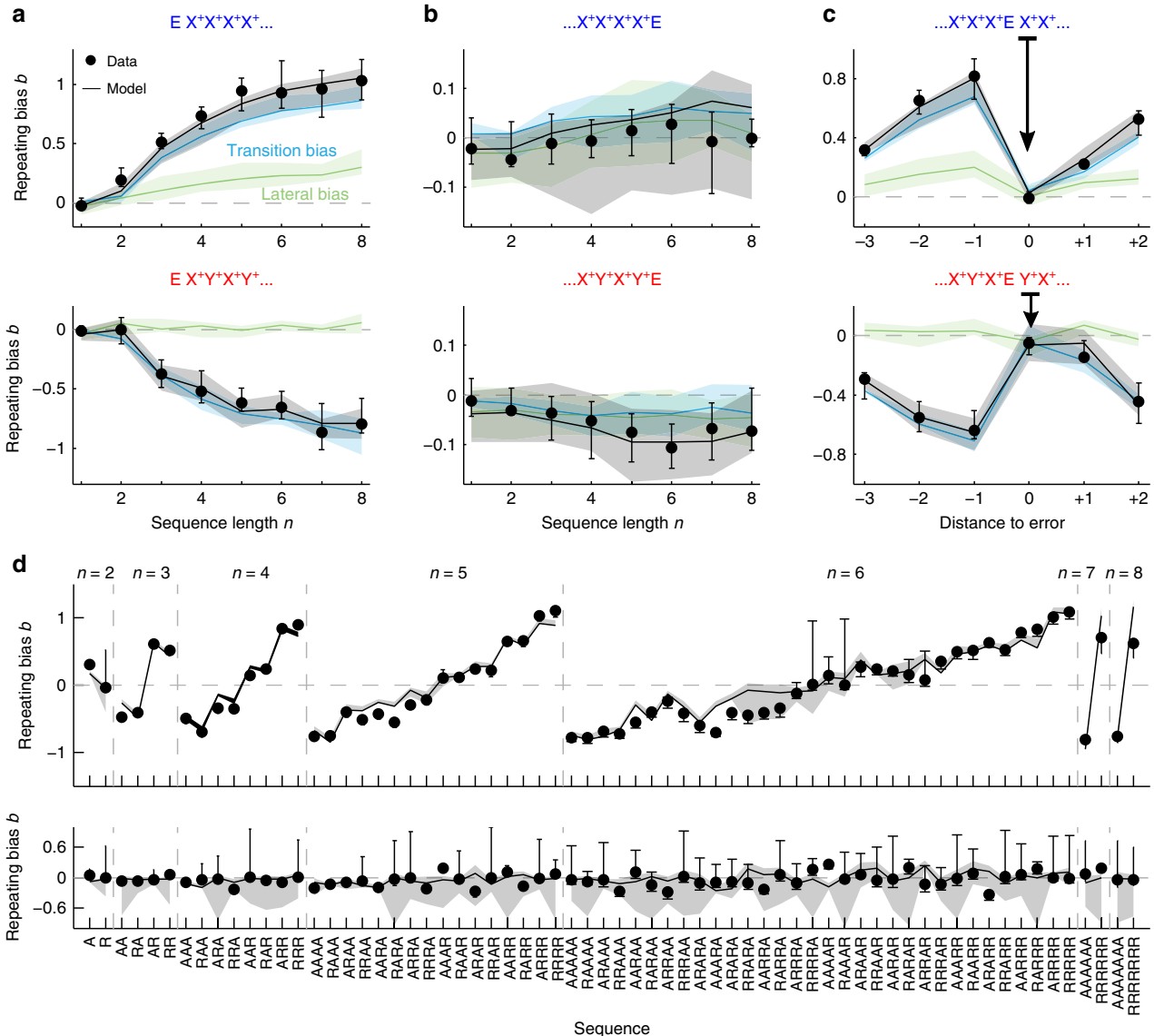

**Fig. 7 Generative model simulation compared to experimental data.** Comparison between experimental data (dots) and model simulation (black curves) showing the repeating bias $b$ for different trial sequences. In the model, $b$ was decomposed into the transition bias contribution (blue curves) and lateral bias contribution (green curves). **a** Repeating bias $b$ versus number $n$ of correct repetitions (top) or alternations (bottom). Data are the same as the color curves in Fig. 2e. **b** Repeating bias versus $n$ after a repetitive (top) or alternating sequence (bottom) terminated by an error (as black curves in Fig. 2e). Notice the different range in the $b$ axes compared with **a**. **c** Repeating bias for sequences with an error E flanked by correct repetitions (top) or alternations (bottom). The bias $b$ is given as a function the trial distance to the error response (distance zero represents $b$ after the error). **d** Repeating bias for all sequences made of $n \leq 8$ repetitions (R) and alternations (A). Top panel shows correct sequences while bottom panel shows correct sequences terminated by an error. In all panels, data and model show median across $n = 10$ rats. Error bars and shaded areas show first and third quartiles.

very good fit to $b$ for all possible sequences of 2–6 correct trials (Pearson's $r = 0.96$; Fig. 7d). Finally, subsequent fitting of our GLM to the simulated data showed that the dynamical model was able to reproduce the history kernels displayed by the animals behavior (Supplementary Fig. 16). In sum, by factorizing the transition bias into accumulated transition evidence $z^T$ and the modulatory signal $c^T$, the model captured the nonlinear across-trial dynamics of history-dependent biases pointing toward possible modulatory circuit mechanisms that could implement this computation (see "Discussion").

## Discussion

We employed a standard acoustic 2AFC task to characterize how rats' perceptual categorizations are affected by expectations

derived from the history of past stimuli, choices, and outcomes, and how these expectations can be captured by a simple dynamical model. A thorough analysis of the behavior isolated two main sequential effects. First, we identified a sequential lateral effect that biased choices towards or away from the recently rewarded or unrewarded targets, respectively (Fig. 4a). This win-stay-lose-switch strategy has been extensively characterized both in humans[5,7,10] and rodents[12,14]. Second, we identified the sequential transition bias, a form of rule bias that had been previously shown to impact human reaction times[8,30,31], choices[32], and neural responses[25,33]. Our results, however, go beyond previous reports in several important aspects regarding error responses: first, repetitions or alternations did not influence subsequent choices whenever one of the two trials of the

transition was unrewarded, meaning that the running estimate of the transition probabilities only accumulated evidence from repetitions and alternations of two rewarded trials. Second, the transition bias was reset after an error trial, i.e., animal responses temporarily ignored the recent history of rewarded repetitions and alternations. However, this reset did not imply the reset of the accumulated transition evidence, i.e., the tally keeping track of the number of recent repetition vs. alternations, whose influence over behavior was restored as soon as the animal obtained a reward (Fig. 5c).

The probability of a subject to repeat the previous response (Fig. 7a) is a common measure to characterize history effects[5,34]. By dissecting the distinct contribution of both first- and second-order serial biases[11,25,35,36], i.e., the lateral and the transition biases, respectively, to the repeating bias we were able to understand the asymmetry in its magnitude between the repeating and alternating blocks (Fig. 2c–e): in series of correct repetitions, both transition and lateral bias add up and yield a strong tendency of the animals to repeat the last rewarded response (Fig. 7a, top). In contrast, in alternating environments, the lateral bias does not build up and the negative repeating bias (tendency to switch) is solely given by the transition bias (Fig. 7a, bottom). In sum, the first-order lateral bias favors repetition over alternation; the second-order transition bias has a symmetric effect. In fact, our analysis provides indirect evidence that animals recapitulated previous repetitions and alternations into a single- and symmetric- transition bias and not into separate variables (Supplementary Fig. 8c). A recent modeling study has proposed that estimating first- and second-order rates is part of the same core computation that the brain performs when analyzing binary sequences[11]. This computation comes down to estimate the two independent transition probabilities $p(r_t | r_{t-1} = -1)$ and $p(r_t | r_{t-1} = +1)$ between consecutive trials $t-1$ and $t$. Our findings seem at odds with this hypothesis because the dependence of each type of bias on the response outcome was very different: whereas incorrect responses $r^-$ tended to cause a negative switch effect (Fig. 4a), incorrect transitions ($T^{+-}$, $T^{-+}$, and $T^{--}$) had no effect (Fig. 4b). Furthermore, only the transition bias showed a reset-and-rebound dynamics caused by error responses (Fig. 5c; Supplementary Fig. 6f–h). Moreover, only the lateral bias showed a dependence on the length of the previous intertrial interval (ITI)[37]: while both after-correct and after-error lateral biases became more positive for longer ITIs (i.e., favoring more repetitions), the transition bias remained unaffected (Supplementary Fig. 2d). An alternative hypothesis, based on analysis of response evoked potentials (ERP), proposes that the lateral bias is generated by the processing of the response, whereas the transition bias from the stimulus processing[25,35]. Preliminary data obtained in the same task in the absence of any stimuli seems to indicate that the transition bias is still present and thus does not seem to be contingent on the processing of sensory inputs.

Several of our findings, together with previous literature[8,25,30,31,33], suggest that the transition bias is a fundamental aspect of sequence processing preserved across subjects, species, and conditions, and which does not seem particularly adaptive to the details of the experiment. First, the transition bias was the same in both repeating and alternating blocks (Supplementary Fig. 9) reflecting the use of a single fixed strategy that could switch from generating a net positive repeating bias in a Repeating block to generating a negative bias in the alternating block (Fig. 2e). Interestingly, this invariance of the transition bias across the repetitive and alternating blocks has also been found in humans performing a 2AFC task[25]. Second, the transition bias was also present when sequences are uncorrelated and the bias can only hinder performance[8,25,30,31,33] (Supplementary Fig. 12c). Third, the trial integration window over which animals estimated the repetition rate (~3–5 trials; Fig. 4b) does not seem adapted to the

block length (200 trials). This short-span estimate allowed to reverse the repetition bias rapidly after a block switch (Supplementary Fig. 1a) at the cost of a noisier estimate of the repetition rate[21,38,39]. Quantification of this integration window in human subjects performing different 2AFC tasks yields numbers in the range of 2–10 trials, despite the use of very long trial blocks with constant sequence correlations[25]. Thus, rather than an overestimation of the environment's volatility[38,40], the short fixed windows might reflect structural adaptation to the statistics of natural environments[41] or a capacity limitation of the system. Fourth, the sophisticated outcome-dependent across-trial dynamics of the transition bias were found systematically in every animal we tested (Fig. 4c) showing that they do not reflect idiosyncratic strategies but the action of an unknown basic cognitive process. Finally, there was one aspect of the mechanism that seemed adaptive: the magnitude of the transition kernel gradually increased when animals, initially trained using uncorrelated sequences, were presented with correlated sequences (Supplementary Fig. 12). Rats in fact had been previously shown to suppress sequential biases when those can be turned up against them[20]. Thus, the transition bias can be adapted to the temporal structure of the environment, if not in nature, at least in magnitude[5] (Supplementary Fig. 12).

Why does the transition bias reset after errors? The question is relevant because an ideal observer who was able to infer the location of the reward in correct and error trials would not show any reset[4,11] (Supplementary Fig. 17). Nevertheless, previous studies have shown that an uncued change in stimulus–outcome contingencies leading to an unexpected number of unrewarded choices can trigger an abrupt behavioral change in rats, switching from the exploitation of a statistical model of the environment to an exploration mode, in which they sample the environment in an unbiased way in order to build new beliefs[18]. This suggests that the reset-and-rebound dynamics of the transition bias could be interpreted as a fast switching between the exploitation of their internal model, represented by their estimate of the transition probability, to a mode that relies almost exclusively on sensory information. This expectation-free mode, however, is different from the standard exploration mode in which animals guide their choices aiming to reduce the uncertainty of the environment. In contrast, our animals, perhaps unable to use their prior after not obtaining the reward (i.e., not knowing what they must repeat/ alternate after an error), guide their choices based on the sensory evidence alone. To capture the reset-and-rebound dynamics, we built a generative-sufficient novel model that could jointly describe the latent trial-to-trial dynamics of (1) the expectation formation following standard reinforcement learning updating rules[40] (Fig. 6a–e) and (2) a modulatory signal $c^T$ that had a multiplicative effect on the impact of the transition evidence in biasing choices. The fitting of the model parameters revealed that $c^T$ reset to zero after errors and then increased progressively with a series of correct trials (Fig. 6f). This modulatory variable may reflect subjects' confidence in their internal model of the environment statistics or, alternatively, the probability that the subject operated in the exploitation mode versus the expectation-free mode. Furthermore, in this expectation-free mode in which the prior is not used, it also cannot be updated with new transition information, as can be concluded from the finding that only ++ transitions impacted subsequent choices (Fig. 4b).

Previous studies on history biases during perceptual tasks in humans found that, in the absence of feedback, the impact of a choice on the subsequent trial was weaker if the subject was unsure of her choice[7,22,23]. The explanation provided in two of these studies was that, according to a normative theory describing how to accumulate noisy evidence in the face of uncued changes[42], low confidence choices should have a weaker contribution

on the belief about what will be the next stimulus category[7]. In our latent variable model, this is indeed true because unrewarded transitions, $T^{+-}$ and $T^{--}$, supposedly generating the lowest confidence about what the true transition was, have a weaker contribution to the accumulated evidence $z^T$ (see fitted values of $\Delta_T$ in Fig. 6d). However, a bias reset after incorrect or low confidence trials was not reported in these studies, i.e., errors without feedback did not seem to modulate retroactively the impact of previous trials onto the next choice, unlike what was observed in our rats. Also, in ref. [7], subjects were informed about the existence of "more repeating", "more alternating", and "uncorrelated" sessions. In contrast, our animals were constantly estimating the transition probability, which varied in blocks during each session. A new study using a two-person matching-pennies game that provided feedback about the choice outcome, found that, after correct responses, subjects responded using a recent history transition bias, "but reverted to stochastic [history-independent] selection following losses"[43] (i.e., errors). Whether this stochastic mode represents the gating off of the transition bias remains to be elucidated. However, the similarity of the two results suggests that a key feature to reproduce the expectation bias reset in future experiments is the use of trial-to-trial feedback rather than fine-tuning other aspects of the task (e.g., sensory stimuli, particular correlation structure in the sequence of stimuli, etc).

The activation of noradrenergic inputs onto the anterior cingulate cortex has been shown to control the switching into a behavioral mode, in which beliefs based on previous experience do not guide choices[20]. Because in the quoted study the experimental condition was a free choice task, removing the impact of history effects resulted in stochastic exploration[20]. This prompts the question of whether the activation of the very same modulatory pathway underlies the after-error switch into the expectation-free sensory-based mode observed in our task. Future pharmacological an electrophysiological experiments will shed light into the brain regions encoding the expectation signals, their modulatory variables as well the circuit mechanisms underlying their combination with the incoming sensory information.

## Methods

All experimental procedures were approved by the local ethics committee (Comité d'Experimentació Animal, Universitat de Barcelona, Spain, Ref 390/14).

**Animal subjects.** Animals were male Long-Evans rats ($n = 25$, 350–650 g; Charles River), pair-housed during behavioral training and kept on stable conditions of temperature (23 °C) and humidity (60%) with a constant light–dark cycle (12 h:12 h, experiments were conducted during the light phase). Rats had free access to food, but water was restricted to behavioral sessions. Free water during a limited period was provided on days with no experimental sessions.

**Task description.** The two tasks performed were auditory reaction-time two-alternative forced choice procedures: an LED on the center port indicated that the rat could start the trial by poking in (Fig. 1a). After a fixation period of 300 ms, the LED went off and an acoustic stimulus consisting in a superposition of two amplitude-modulated sounds (see details below) was presented. The rats had to discriminate the dominant sound and seek reward in the associated port. Animals could respond any time after stimulus onset. Withdrawal from the center port during the stimulus immediately stopped the stimulus. Correct responses were rewarded with a 24 μl drop of water, and incorrect responses were punished with a bright light and a 5 s timeout. Trials in which the rat did not make a side poke response within 4 s after leaving the center port were considered invalid trials and were excluded from the analysis (on average, only 0.4% of the trials were invalid). Behavioral setup (Island Motion, NY) was controlled by a custom software developed in Matlab (Mathworks, Natick, MA), based on the open-source BControl framework (http://brodylab.princeton.edu/bcontrol). Rats performed an average of 694 trials per session (range: 335–1188), one session per day lasting 60–90 min, 6 days per week, during 9 months. Rats were trained using an automated training protocol that had several stages and lasted between 2 and 3 months (depending of the animal). The data presented in this study were taken from the period after training yielding an average of 56,506 valid trials per rat. A first group of $n = 10$ rats (Group 1) were trained in the frequency discrimination version of the task, in which the correlated sequence of trials was present from the training

(see below). A subset of three rats from this group were also trained in a random time-out version of the task where the duration of the after-error timeout was randomly chosen between 1, 3, or 5 s. A second group of $n = 9$ rats (Group 2) were trained in a level discrimination version of the task using the same correlated sequence than the first group. A third group of $n = 6$ rats (Group 3) were trained in the frequency discrimination version of the task but starting with uncorrelated stimulus sequences and only after several weeks, introducing the correlated sequences used in the first group of animals.

**Acoustic stimulus.** In the two acoustic tasks used, the stimulus $S_k(t)$ in the $k$th trial was created by simultaneously playing two amplitude-modulated (AM) sounds $T_R(t)$ and $T_L(t)$:

$$S_k(t) = [1 + sin(f_{AM}t + \varphi)][a_k^L(t)T_L(t) + a_k^R(t)T_R(t)] \quad (1)$$

The AM frequency was $f_{AM} = 20$ Hz, and the phase delay $\varphi = 3\pi/2$ made the envelope zero at $t = 0$. In the frequency discrimination task, $T_L(t)$ and $T_R(t)$ were pure tones with frequencies 6.5 kHz and 31 kHz, respectively, played simultaneously in the two speakers. In the level discrimination task (Supplementary Fig. 11), they were broadband noise bursts played on the left and on the right speaker, respectively. The amplitudes of the sounds $T_L(t)$ and $T_R(t)$ were separately calibrated at 70 dB. Sounds were delivered through generic electromagnetic dynamic speakers (STAX, SRS-2170) located on each side of the chamber, and calibrated using a free-field microphone (Med Associates Inc, ANL-940-1).

**Stimulus sequence.** The Markov chain generated the sequence of stimulus category $c_k = \{-1,1\}$, that determined whether the reward in the $k$th trial was available in the left or the right port, respectively (Fig. 1b, top). The stimulus category $c_k$ set which of the two sounds, $T_L(t)$ and $T_R(t)$, composing each stimulus was dominant this ultimately determined the statistics of the sound amplitudes $a_k^L(t)$ and $a_k^R(t)$ (Eq. 1) as described below. In each trial, independently of $c_k$, the stimulus strength $s_k$ was also randomly generated (Fig. 1b, bottom). Stimulus strength $s_k$ defined the relative weights of the dominant and nondominant sounds: for example, when $s_k = 1$ only the dominant sound was played (i.e., easiest trials), whereas when $s_k = 0$ the two sounds had on average the same amplitude (i.e., hardest trials). We used four possible values for $s = 0, 0.23, 0.48$, and 1. The stimulus evidence was defined in each trial as the combination $e_k = c_k*s_k$. The value of $e_k$ determined the p.d.f. from which the instantaneous evidence $S_{k,f}$ was drawn in each frame $f$ (i.e., in each 50 ms AM-envelope cycle; Fig. 1d, top): when $e_k := \pm 1$ the p.d.f. was $f(x) = \delta(x \mp 1)$ (i.e., a Dirac delta p.d.f.), whereas when $e_k \in (-1,1)$, it was a stretched beta distribution with support $[-1,1]$, mean equal to $e_k$ and variance equal to 0.06 (Fig. 1d, top). Finally, the amplitudes $a_k^L(t)$ and $a_k^R(t)$ of the two AM envelopes (Eq. 1) were obtained using $a_k^L(t) = \left(1 + S_{k,f}\right)/2$ and $a_k^R(t) = \left(1 - S_{k,f}\right)/2$ with $f$ referring to the frame index that corresponds to the time $t$ (see example in Fig. 1d). With this choice, the sum of the two envelopes was constant in all frames $a_k^L(t) + a_k^R(t) = 1$.

**Psychometric curve analysis.** We computed two types of psychometric curves for each animal, by pooling together trials across all sessions for each type of block and for each of the seven different stimulus evidences ($e = 0, \pm0.23, \pm0.48, \pm1$). We calculated (1) the proportion of rightward responses vs. stimulus evidence $e$ (Fig. 1a, left) and (2) the proportion of repeated responses as a function of the repeating stimulus evidence $\hat{e}$ defined for the $t$th trial as $\hat{e}_t = r_{t-1}e_t$, with $r_{t-1} = \{-1,1\}$ representing if the response in the previous trial was left or right, respectively (Fig. 1b). Thus, positive (negative)-repeating stimulus evidence denotes trials, in which the animals had evidence to repeat (alternate) their previous choice. In other words, a rightward stimulus with evidence $e_t = 0.23$ after a left response implied a repeating stimulus evidence equal to $\hat{e}_t = -0.23$. Both psychometric curves were separately fitted to a 2-parameter probit function (using Matlab function *nlinfit*):

$$P_{\text{Rightwards}}(e) = \frac{1}{2}\left(1 + erf\left(\frac{\beta e + b}{\sqrt{2}}\right)\right) \quad (2)$$

$$P_{\text{Repeat}}(\hat{e}) = \frac{1}{2}\left(1 + erf\left(\frac{\beta'\hat{e} + B}{\sqrt{2}}\right)\right). \quad (3)$$

The sensitivities $\beta$ and $\beta'$ quantified the stimulus discrimination ability, while the fixed side bias $B$ captured the animal side preference for the left ($B < 0$) or right port ($B > 0$), and the repeating bias $b$ captured the animal's tendency to repeat ($b > 0$) or alternate ($b < 0$) their previous choice. Within-subject error bars were estimated by one standard deviation of a nonparametric bootstrap ($n = 1000$). Across-subject error bars, corresponded to the 1st and 3rd quartiles.

**Generalized linear model (GLM) analysis.** We built a GLM where different features, such as the current stimulus and previous history events, were linearly summed to give rise to the probability that the rat's response $r_t$ in trial $t$ was toward

the right port[7,10,12,13,15,28,44]:

$$p(r_t = +1 | \omega, \pi, \beta) = \pi_R + (1 - \pi_L - \pi_R)\Phi(y_t). \quad (4)$$

In which $\pi_R$ and $\pi_L$ represent the lapse rates for left and right responses and $\Phi$ is the cumulative of the standard normal function and its argument reads:

$$y_t = \sum_f \omega_f^S S_{t,f} + \sum_{k=1}^{6} \omega_k^A S_{t-k}^{sum} + \sum_o \sum_{k=1}^{6} \omega_{k,o}^L r_{t-k}^o + \left( \sum_{o,q} \sum_{k=1}^{6} \omega_{k,o,q}^T T_{t-k}^{o,q} \right) r_{t-1} + \beta. \quad (5)$$

The current stimulus was given by $S_{t,f}$ defined as the intensity difference between the two tone sounds in frame $f$ (for $f = 1, 2 \ldots 8$). For the history-dependent contributions, we included the impact of the previous ten trials (i.e., $t-1$, $t-2$,… $t-(6-10)$; we grouped the impact of trials $t-6$ to trial $t-10$ in one term). The impact of previous stimuli was represented by $S_t^{sum} = \sum_f S_{t,f}$. The terms $r_{t-k}{}^+$ represented the previous rewarded ($o = +$) responses being $-1$ (correct left), $+1$ (correct right), or 0 (error response). Similarly, $r_{t-k}{}^-$ represented previous unrewarded ($o = -$) responses being $-1$ (incorrect left), $+1$ (incorrect right), or 0 (correct response). Previous transitions were given by $T_{t-k}^{o,q} = r_{t-k-1}^o r_{t-k}^q$ and were separated into $\{o,q\} = \{+, +\}$, $\{+, -\}$, $\{-, +\}$, and $\{-, -\}$, depending on the outcomes of trial $t-k$ (q) and $t-k-1$ (o). $T_{t-k}^{++}$ transitions, for example, were $+1$ for repetitions between correct responses, $-1$ for alternations, and 0 for when either of the responses was an error. The parameter $\beta$ captured a fixed side bias. The sets of weights $\omega_f^S$, $\omega_k^A$, $\omega_{k,o}^L$, and $\omega_{k,o,q}^T$ were fitted, together with the parameters $\pi_R$, $\pi_L$, and $\beta$, to the responses of each rat separately using a generalized expectation–maximization algorithm implemented in Matlab[10] (see section 2 in Supplementary Methods for details).

**Dynamic variable model of behavior**. We developed a dynamical model of the rats behavior in which three latent variables $z^L$ and $z^T$ accumulated and maintained the lateral and transition evidence, respectively. These variables were updated in each trial using standard updating rules that accumulated responses $r_t^o$ or transitions $T_t^{o,q}$ with weights $\Delta_L^o$ and $\Delta_T^q$, respectively, and had leak terms $\lambda_L^o$ and $\lambda_T^q$ that were outcome dependent ($o, q = +, -$). The variable $c^T$ was a variable that modulated the impact of $z^T$ on choice and was updated on each trial based only on the trial's outcome. The model combined linearly the net stimulus evidence, the lateral bias $\gamma^L = z^L$ and the transition bias $\gamma^T = c^T \times z^L \times r_{t-1}$ and passed them through a probit function. The parameters of the model were fitted by maximizing the log posterior of the observed responses of each rat using the function *fmincon* from the Matlab Optimization Toolbox (see section 3 in Supplementary Methods for details).

**Reporting summary**. Further information on research design is available in the Nature Research Reporting Summary linked to this article.

## Data availability
The data generated during the study are available in a public repository (https://osf.io/mktdb/).

## Code availability
The codes generated during the study are available in a public repository (https://osf.io/mktdb/).

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

## Acknowledgements

We thank Anke Braun, Anne E. Urai, Daniel Duque, Gabriela Mochol, Genís Prat-Ortega, Lluís Hernández, Manuel Molano, Tobias H. Donner, and Yerko Fuentealba for excellent discussions and critical reading of the paper, and Lejla Bektic for help with training of the animals. This research was supported by the Spanish Ministry of Economy and Competitiveness together with the European Regional Development Fund (BES-2011-049131 to A.H.M. and SAF2015-70324-R to J.R.). This work has received funding from the European Research Council (ERC) under the European Union's Horizon 2020 research and innovation programme (Grant agreements No. 683209 to J.R. and 615699 to D.R), from the European Union Seventh Framework Programme (Marie Curie auction CEMNET project 629613 to A.H.; Marie Curie IIF253873 to P.R.-O.), from Mexico CONACyT (144335 to P.R.-O.) and the USA National Institute on Deafness and Other Communication Disorders (R01DC015531 to S.J.). Part of this work was developed at the building Centro Esther Koplowitz, Barcelona.

## Author contributions

A.H.M. and J.R. designed the experiments with the assistance from D.R. P.R.-O. and S.J.; A.H.M. carried out all the experiments; A.H.M. and A.H. analyzed the data; A.H. developed the generative dynamical model; A.H.M., A.H. and J.R. interpreted the data and wrote the paper with contributions from the rest of the authors.

## Competing interests

The authors declare no competing interests.
