## [Peer Review File · Nature Communications]

Reviewers' Comments:

Reviewer #1:

Remarks to the Author:

In this paper, Hermoso-Mendizabal et al. administered rats to a perceptual decision-making task, in which subjects could improve their likelihood of making a correct response by having a bias to repeat (in a Repetitive Block) or alternate (in an Alternating Block) their response. The authors found that rats indeed utilized the bias; but interestingly, subjects showed no bias on trials immediately after making errors.

I found the main finding of no bias after error trials very interesting. But I also found their analysis very confusing. I hope that authors can clarify my confusions.

1. The authors first introduced a block-dependent repeating bias b , and a fixed side bias B (page 3). These are very important variables to support the main findings, but I couldn't find how these parameters are defined and how they are estimated. Please describe.

2. The authors performed several model comparisons using BIC scores in order to justify their full model with biases (e.g. page 9). I was confused with this analysis. First, the authors seem to favor larger BIC scores, but a conventional definition of BIC ($-\log$ likelihood + penalty) favors models with smaller BIC scores. But I couldn't find a definition of BIC in the manuscript. Please clarify the definition of BIC.

3. (Related to 2). I hope I'm wrong about the BIC sign confusion. But I am concerned because their full model has so many parameters, and it is usually very difficult to BIC-justify adding free parameters independently to kernels on each trial (separately for $t-1$, $t-2$, $t-3$, ... $t-10$, leading to adding 10 parameters), just as the authors did. The penalty term of BIC is proportional to $\log(\text{number of trials}) \times \text{number of parameters}$. Because rats performed so many trials, changes in this term by adding parameters are likely to be very large. Could you also clarify the number of parameters in each model that went into the BIC score, and discuss how the full model with many parameters is BIC-justified unlike other standard studies?

4. In page 4, the authors state "The reset was only observed after errors and not after correct but unexpected responses, e.g. one alternation after several repetitions (Supplementary Fig. 2a)". But Figure S2a looks as though the biases were gone after unexpected responses (colored lines are significantly different from black lines). So I'm very confused with this contradiction. Could you clarify?

5. It took really long time for me to understand Figure 4. One reason is that the model in Figure 3 does not straightforwardly relate to estimates in Figure 4 (variables in figure 3 and 4 appear to be totally different). If I understood correctly, instead of estimating the parameters of the model shown in figure 3 directly, the authors estimated related parameters shown in Figure 4. Is this right? But then the readers will have a hard time to visualize the mapping between the model in figure 3 and the results in figure 4 (the texts in page 7 are also very complicated...) Also it should be explained why there are missing points in figure 4 (e.g. red at $t=2$ in $r+$). I hope the authors can clarify these points.

6. The authors performed a control experiment with different inter-trial-intervals and found no effects on repeating bias after errors (page 4; Figure S1). This is very interesting, but how about kernels for correct trials? A recent similar study showed that inter-trial-intervals affected decision strategies in mice (Iigaya, Fonseca, Murakami, Mainen, and Dayan, 2018). It would be interesting if the authors could discuss results for correct trials, too.

7. It is presumably good to have repeat/alternate biases in the authors' task. How do the animals' kernels compare with optimal bias kernels for the task, given their performance in perceptual decisions?

Reviewer #2:

Remarks to the Author:

This is an interesting study examining how rats use priors and current sensory cues to make choices using behavioural measurements and computational models. Behavioural tasks are nicely designed and conducted, and the analyses are performed with care. The fact that animals learn to repeat or alternate their choices depending on the block type is not surprising. However, the manuscript reports that immediately after error trials, the effect of priors goes away, and then comes back as soon as animals start to perform correctly. This aspect of the behaviour is interesting and novel.

Given that the task has an active initiation for each trial, one question is whether animals take longer to initiate the trial after an error trial, compared to the other trials. It could be that after an error they take much longer to initiate the next trial, and they then show less effect of trial history. It is necessary to look into animals' initiation time to check for these and similar possibilities.

In addition to initiation time, choice reaction times should be analysed; Whether animals' choice reaction times show the same effect as choices? Could it be that after an error choices are reset but reaction times are not? It would be useful to investigate animals RT, in particular given the large number of trials collected and the fact that the task is a reaction time task.

In this type of experiments, error trials could happen more frequently towards the end of the session, as the motivation slowly drops. If this is the case in these experiments, then it would be necessary to perform analysis which takes this into account and/or repeat plots such as figure 2 only for late trials of the sessions.

Similarly, it is reasonable to think that several of errors are actually happening right after the block switch. Thus, it would be useful to compare error trials and correct trials that happen at the onset of the blocks.

Lastly, I am wondering whether it would make sense to move the first regression section (Figure 4) to the Supplements. I am suggesting this because I found it hard to digest that regression and then follow the main model. This is just a suggestion as authors might find it necessary to have both analyses in the main text.

Minor:

Figure 5C benefits from some legends describing the colours.
Figures 6b-f also benefit from such legends.

Reviewer #3:

Remarks to the Author:

In this manuscript, Hermoso-Mendizabal and colleagues investigate the impact of past-trial influences on auditory perceptual decisions, using a combination of behavioral experiments and various

modelling approaches. Investigating psychometric curves, they first demonstrate that psychometric curves are influenced by the repeat/transition probabilities of the current block of trials, but only after a correct previous trial. To get insights into the specific factors underlying this effect, they then use a GLM, with which they can dissociate impact of previous responses (lateral bias) vs. previous repetitions/alternations (transition bias). Analysis of the fitted coefficients reveals that the effect is driven by previous transitions, but again only if responses were correct. Third, investigating the correlation of transition bias across lags (transfer coefficient), the authors can demonstrate that the transition bias is reset after an error, but re-emerges following a correct response afterwards. These insights are further corroborated by a generative model, which could capture the effects of past stimuli and outcomes on internal variables, but also successfully predicted the animals' biases in simulations.

Overall, the manuscript reveals fundamental insights into overarching principles underlying behavior in a 2AFC task, which will be of general interest to circuits neuroscientists working in animal models, cognitive neuroscientists working in humans, and those interested in behavior.

MAJOR

- Supplementary Fig. 4: I would strongly recommend moving this figure to the main text. The GLM is a very powerfully laid out to not only capture psychophysical kernels, past-trial sensory adaptation effects, but also strategies like "win-stay", "loose-switch". Hiding all of this in the supplementary material not only makes the text hard to follow, but would also be a pity. While the figure is very helpful, several things regarding the GLM remain unclear, also because the text in supplementary material is a bit more raw than for the main text.

* - Supplementary Material, p. 4 "The net stimulus evidence of each of the previous trials ($t-1$, $t-2$, ..., green and purple bars in gray box) are weighted by the previous stimulus kernel". What exactly is "previous stimulus kernel"? In the text, it says: sums of stimulus evidence over all frames, which sounds more straightforward.

* - Supplementary Material, p. 14: description of Lateral module: what is i , j ?

- Can the authors please explain in more detail why they chose to group regressors into specific sums representing the various modules?

- Can the authors please also clarify the use of variance to estimate each module's estimated bias (the beginning of sections 2.2 would be informative even earlier in the text, i.e. after the description of the transition module).

Similar issues apply to Fig. 5 and associated supplementary text: Please move some of the explanation of what a transfer coefficient is from figure legend to proper text, and streamline the associated supplementary text to help the reader find the appropriate information. Section 2.5 of the supplementary material almost starts like a second version of the main text, which is not helpful. The explanation of transfer coefficients (in supplementary material) is not easy to follow and seems still raw.

- Maybe I missed it, but in fitting the GLM or other models, was there some cross-validation involved?

- It might be interesting to compare simulations by the GLM and the generative model to get insights into how performance of these two approaches compare.

MINOR:

- p. 3: "repeating psychometric curves; sensory evidence in favour of the repeating choice": this is impossible to understand based on results alone, and only the example in the methods section really helps. I would encourage the authors to re-phrase and/or expand the explanation in the results session

- 2c: I would encourage the authors to plot the repeating / fixed side bias on the same scale since it

will allow better judging of their relative sizes

- 2e: is the difference between the repeating bias b for repetitions vs. alternations significant? Please modify statement "This plateau was greater after repetitive patterns rather than alternating patterns" to include the statistics. Also, the onset seems different. Can the authors please comment?
- p.4: increased with n (n what? I assume trials)
- p. 4: "The impact on performance was largest for low stimulus strength etc." Can the authors please provide the statistics and explain explicitly how they came to the conclusion?
- p. 7: "This is achieved by multiplying zT with the last response r_{t-1} ," - I would recommend adding here that this does not necessarily have to apply, but happens in the authors' model, i.e. "This is achieved in our framework..." or something along these lines
- p. 7: "Lateral and transition biases have an opposite influence in the final choice" - I would recommend adding "In this example, lateral ..." - they can also have congruent influences, e.g. in the decisions before the final choice
- p. 7: "... since it allows to capitalize the sequence correlations in both types of blocks" - consider rephrasing, this is not entirely clear (what exactly is 'it'? what does 'capitalize' mean in this context?)
- p. 7: see Methods for details -> Supplementary Methods - it took me a while to find...
- p. 8, Figure legend: "Influence of the response side (Left vs Right) from previously rewarded ($r+$, left panel) and unrewarded ($r-$, right panel) trials". Shouldn't this say "Influence ON the response side (Prob Right) OF previously ..."?
- p. 7: "The contribution of each response to γ_L depended on its outcome following a win-stay-lose-switch pattern" - please rephrase, this is unclear.
- Fig. 4a: Maybe I missed this in the main text, but please explain with a quick sentence and reference to supplementary material (Indeterminacy analysis) why there are data missing from certain trial lags.
- Fig. S5 legend: "This repulsive side bias captured the effect of having heard the previous exact stimuli, above and beyond the bias introduced by the category of those stimuli which was captured by the lateral bias" -> previous exact stimuli is unclear; above AND beyond
- p. 15: "go beyond from previous reports" -> go beyond previous
- p. 18: "(see below)" misplaced here
- Supplementary Material, p. 14: "variability on" -> variability of
- Supplementary Material: "Compared with the GLM that required that rats maintained memory of all the features for each of the 10 previous trials in order to make a decision, this latent variable model reduced the working memory load to simply maintaining the value of three latent variables." - This seems odd - the GLM has no influence whatsoever on the rats' behavior.

We would like to start by thanking the three reviewers for their very careful reading of the manuscript and their valuable comments and questions. We think that by addressing them, the paper is now clearer and stronger regarding the main findings. We here address point by point the comments of each referee (rewritten in black), providing explanations and clarifications (in blue) and pasting the new additions and edits to the paper that address these comments (colored in red in this document and the main text).

Besides addressing the reviewers comments, we have added in the Discussion a new reference to a study that was just published showing a very similar form of *transition bias reset* in humans playing a matching pennies game (Kikumoto and Mayr, eLife 2019).

Reviewer #1:

In this paper, Hermoso-Mendizabal et al. administered rats to a perceptual decision-making task, in which subjects could improve their likelihood of making a correct response by having a bias to repeat (in a Repetitive Block) or alternate (in an Alternating Block) their response. The authors found that rats indeed utilized the bias; but interestingly, subjects showed no bias on trials immediately after making errors.

I found the main finding of no bias after error trials very interesting. But I also found their analysis very confusing. I hope that authors can clarify my confusions.

1. 1. The authors first introduced a block-dependent repeating bias b , and a fixed side bias B (page 3). These are very important variables to support the main findings, but I couldn't find how these parameters are defined and how they are estimated. Please describe

We have now expanded the explanation of this important Repeating Psychometric Curve in the Results section (p. 3):

“The horizontal shift of this psychometric curve, parametrized by the fixed side bias B , measured the history-independent subject's preference towards one side. To estimate the impact on choice of the serial correlations of the stimulus sequence, we also analyzed the *repeating psychometric curve*, showing the proportion of trials where the animals repeated the previous choice as a function of the sensory evidence favoring to repeat the previous choice (see Methods for details). The horizontal shift of this new psychometric curve, parametrized by the repeating bias b , measured the history-dependent tendency to repeat or alternate the previous choice.”

We have also made the definition of B and b in the Methods more clear by including the following paragraph (p. 20-21):

“We calculated (1) the proportion of Rightward responses vs. stimulus evidence e (Fig. 1a left) and (2) the Proportion of Repeated responses as a function of the Repeating Stimulus Evidence \hat{e} defined for the t -th trial as $\hat{e}_t = r_{t-1}e_t$, with $r_{t-1} = \{-1, 1\}$ representing if the response in the previous trial was Left or Right, respectively (Fig. 1b). Thus positive (negative) Repeating Stimulus Evidence denote trials in which the animals had evidence to repeat (alternate) their previous choice. In other words, a Rightward stimulus with evidence

$e_t = 0.23$ after a Left response implied a repeating stimulus evidence equal to $\hat{e}_t = -0.23$. Both psychometric curves were separately fitted to a 2-parameter probit function (using Matlab function *nlinfit*):

$$P_{Rightwards}(e) = \frac{1}{2} \left(1 + \operatorname{erf}\left(\frac{\beta e + b}{\sqrt{2}}\right) \right) \quad (2)$$

$$P_{Repeat}(\hat{e}) = \frac{1}{2} \left(1 + \operatorname{erf}\left(\frac{\beta \hat{e} + B}{\sqrt{2}}\right) \right) \quad (3)$$

The sensitivities β and β' quantified the stimulus discrimination ability, while the Fixed side bias B captured the animal side preference for the Left ($B < 0$) or Right port ($B > 0$), and the repeating bias b captured the animal's tendency to repeat ($b > 0$) or alternate ($b < 0$) their previous choice."

1. 2. The authors performed several model comparisons using BIC scores in order to justify their full model with biases (e.g. page 9). I was confused with this analysis. First, the authors seem to favor larger BIC scores, but a conventional definition of BIC (-log likelihood + penalty) favors models with smaller BIC scores. But I couldn't find a definition of BIC in the manuscript. Please clarify the definition of BIC.

We thank the reviewer for pointing this important lack. Two different and conflicting definitions for the BIC are usually used. The one described by the reviewer (-2log likelihood + penalty), meant as a deviance measure, indeed favouring models with smaller BIC scores. The other definition, provided in the original formulation of BIC by G. Schwarz (*Annals of Statistics, vol 6, p461-2, 1978*), is log-likelihood - penalty/2 (i.e. -1/2 times the other definition), derived as an approximation to the log model evidence. It thus favors models with larger BIC scores. That is also the definition provided in the Bishop machine learning handbook, and its derivation is explained here: http://wittawat.com/posts/laplace_apporx_bic.html.

We have used this second somehow-less-usual definition, and thus favor larger BIC scores. We are sorry for the confusion and have tried to make it clearer in the manuscript: We have added a section in Supplementary Material to explain this definition, making apparent the contrast with the other usual definition (p. 23 of the Supplementary Methods 2.5 section).

" 2.5 Model comparison

We used the Bayesian Information Criterion (BIC) to compare how well different models (GLMs and dynamical models) accounted for rat behavior. We used the original definition of BIC by Schwartz (add ref) as an approximation to the model evidence:

$$\text{BIC} = \log L(\hat{\theta}) - d/2 \log(n)$$

where $L(\hat{\theta})$ represents the likelihood of the model at the estimated parameters (weights and lapses in the case of the GLM) and d is the number of parameters. That definition contrasts with a common definition scaling negatively with the likelihood. Here, a larger BIC is an indication of a better model."

In legend of Supplementary Figure 8a (p. 7 Supplementary Material), we have also specified:

“We used the BIC definition as penalized log-likelihood (Schwarz,1978), i.e. larger BIC score means better model (see Section 2.5).”

1. 3. (Related to 2). I hope I'm wrong about the BIC sign confusion. But I am concerned because their full model has so many parameters, and it is usually very difficult to BIC-justify adding free parameters independently to kernels on each trial (separately for t-1, t-2, t-3, ...t-10, leading to adding 10 parameters), just as the authors did. The penalty term of BIC is proportional to $\log(\text{number of trials}) \times \text{number of parameters}$. Because rats performed so many trials, changes in this term by adding parameters are likely to be very large. Could you also clarify the number of parameters in each model that went into the BIC score, and discuss how the full model with many parameters is BIC-justified unlike other standard studies?

Again, we are very sorry for the confusion due to not providing the definition of BIC. The BIC is derived as an approximation to the marginal model evidence: while it is more conservative than the AIC due to its stronger penalty for the number of parameters, it will converge to select the better model as the sample size increases. This is because, if a model A provides an *expected* increase of Δ for the likelihood *for each trial* over model B, then the increase for the total log-likelihood will scale as $\text{number of trials} \times \log(\Delta)$. That linear increase clearly wins over the logarithmic penalty as number of trials increase, so it is actually expected that our BIC will select the best model for large sample size despite the penalty (even more reliably than for smaller sample size, which is expected for a statistical estimator). So our interpretation is simply that the full GLM model provides a better account of experimental data than reduced models. This is also consistent with the fact that fitted weights for the full model very clearly depart from 0, indicating new regressors help explain more of animal choices.

That said, we did seek to reduce the number of parameters by for instance grouping the impact of trials at lags 6 to 10 (see e.g. Fig. 4).

1. 4. In page 4, the authors state “The reset was only observed after errors and not after correct but unexpected responses, e.g. one alternation after several repetitions (Supplementary Fig. 2a).” But Figure S2a looks as though the biases were gone after unexpected responses (colored lines are significantly different from black lines). So I'm very confused with this contradiction. Could you clarify?

What we meant here is that, in contrast with the reset to $b=0$ observed after errors, unexpected correct responses decreased the bias b but the bias did not vanish in general. The referee is correct when saying that the repeating bias after a correct *expected* response (i.e. a continuation of the repetitive or alternating pattern) is different than a correct *unexpected* response (i.e. a violation of the pattern). But that is not the comparison we are referring to. We are referring to the comparison of the after-error reset curves (black in Fig. 2e) and the after-correct-unexpected grey curves in the Supplementary Fig. 3a. In order to make this comparison more clear, we have included in Supplementary Fig. 3a the after-error reset curves from Fig. 2e (black curves). For short sequence lengths ($n = 2-4$), the bias b for unexpected correct responses (grey curve) is different than for after-error responses (black curve; paired t -test $p < 0.05$; see gray horizontal gray line in Supp. Fig. 3a). For longer lengths $n > 5$, both curves hover around zero. Our model indeed predicts

that in both blocks the repeating bias after correct-unexpected responses vanishes for $n > 5$ (Supplementary Fig. 3b). This is because the transition bias has the opposite contribution to the lateral bias, so their sum is approximately zero.

We paste here the modified Supplementary Fig. 3 and highlight the changes introduced in the legend to clarify this point (**bright red** marks the newly added text):

Supplementary Figure 3. Repeating bias after unexpected correct responses. a, Repeating bias b versus the number of correct previous repetitions in Repetitive block (blue) or alternations in Alternating block (red). Gray dots show b after a correct unexpected alternation following several repetitions (left) or a correct unexpected repetition following several alternations (right). Black dots show b after an error following several repetitions (left) or following several alternations (right). Breaking the sequence pattern with a correct unexpected response did not bring the repeating bias to zero independently of the sequence length, n as error responses did (compare gray and black dots). The bottom gray line indicates the sequence length n in which the bias after error responses versus after correct unexpected responses were significantly different ($p < 0.05$, paired t-test). **b,** Comparison of the repeating bias b after an unexpected correct response obtained from the experimental data (gray dots; same as gray dots in **a**) and simulations of the full latent variable model (gray curves). In the model, b was decomposed into the contributions of the transition bias (blue curves) and the lateral bias (green curves). Notice that the full model predicts that b goes to zero for large n as a result of the cancelation of these two contributions. Dots show median across $n = 10$ animals (Group 1). Error bars and shaded area show the 1st and 3rd quartiles.

1. 5. It took really long time for me to understand Figure 4. One reason is that the model in Figure 3 does not straightforwardly relate to estimates in Figure 4 (variables in figure 3 and 4 appear to be totally different). If I understood correctly, instead of estimating the parameters of the model shown in figure 3 directly, the authors estimated related parameters shown in Figure 4. Is this right? But then the readers will have a hard time to visualize the mapping between the model in figure 3 and the results in figure 4 (the texts in page 7 are also very complicated...)

The referee is making a valid point. The transition between the conceptual model in Fig. 3 and the GLM model in Fig. 4 was not properly done. We think that the explanation of the difference between the lateral and transition biases is fundamental for the correct understanding of our most important results and it has not been easily described in the literature (i.e. there are no references we could use that illustrate this difference). This is what Fig. 3 is providing.

The GLM is a standard and powerful description of the factors impacting choices in 2AFC tasks with a minimal number of assumptions (i.e. almost model-free). It has been widely used in the field (see e.g. Corrado et al., 2005; Lau and Glimcher, 2005; Abrahamyan et al., 2016; Braun et al., 2018; Busse et al., 2011; Fründ et al., 2014; Nogueira et al., 2017; Urai et al., 2017). Applied to our data, the GLM provides very strong evidence of the existence of the reset of the transition bias. For this reason we think that both the conceptual model (Fig. 3) and the GLM model (Fig. 4) should go in the main text, but we agree that the linkage between the two had to be improved.

We have now modified Fig. 3 and the text presenting Fig. 4 so that the link between these two figures is smoother. This has been done by describing in a bit more detail what the GLM does (i.e. calculates the weights of the different factors) and how this weighting appears in the conceptual model in Fig. 3:

Here is the new version of Fig 3 with its caption:

Figure 3. Dissecting two different history choice biases. Cartoon of an example series of four choices, $R^+R^+R^+L^+$, illustrating the build-up of the lateral and transition biases. **a**, The lateral bias, capturing the tendency to make Rightward or Leftward responses, increases towards the Right in the first three R^+ trials and compensates this build-up with the last L^+ response. Its net impact on the final trial is a Rightward bias. **b**, Schematic of the sequence of rewarded rat responses showing the transitions, defined as the relation between two consecutive responses, being Repetitions (Rep, blue arrows) or Alternations (Alt, red arrow). The animal computes each choice from combining its expectation based on a weighted sum of previous alternations (bottom gray balloon) and previous responses (upper gray balloon) with the current stimulus sensory information (see last trial). **c**, Transition evidence z^T captures the tendency to repeat or alternate the previous response based on the series of previous transitions $Rep^+Rep^+Alt^+$ predicting a Repetition in the final trial. **d**, The transition bias γ^T is obtained by projecting the Transition evidence z^T (c) onto the Right-Left choice space via a multiplication with the previous response r_{t-1} (see gray arrow). **e**, The evidence provided by the current stimulus is summed to the addition of the biases $\gamma_{t-1}^T + \gamma_t^T$ and passed through a sigmoid function, yielding the probability of selecting a Rightward response (Supplementary Fig. 5).

Here is the paragraph in the Results (p. 8) that transitions from the model in Fig 3 to the GLM model:

“To quantitatively assess how subjects computed these biases we used an explorative approach that assumed that, in each trial, animals combined linearly the responses r ($r = R, L$) and transitions T ($T = Rep, Alt$) from the last ten trials in order to generate the lateral and transition bias, respectively (see Fig. gray boxes in Fig. 4c and Supplementary Fig. 5; see Supplementary Methods section 2 for details). The two biases were then combined with the stimulus evidence in order to yield a decision (Fig. 4e). Fitting such a generalized

linear model (GLM) to the behavior of a rat implied finding the weight with which each of these past (e.g. previous transitions) and current events influenced the animal choices (Abrahamyan et al., 2016; Braun et al., 2018; Busse et al., 2011; Fründ et al., 2014; Nogueira et al., 2017; Urai et al., 2017). “

1.5.2. Also it should be explained why there are missing points in figure 4 (e.g. red at t=2 in r+). I hope the authors can clarify these points.

The explanation about this missing points was provided in the Supp Methods (section 2.3). It boils down to the fact that there are colinearities between some of the coefficients in the GLM, meaning that it is not possible to estimate all the weights. In the Supp Methods we explain the criteria to resolve this ambiguity.

We have now added a sentence in the caption of Fig. 4 saying (p. 9):

“See Supplementary Material Section 2.3 for why some weights are missing at trial lags 1 and 2.”

1. 6. The authors performed a control experiment with different inter-trial-intervals and found no effects on repeating bias after errors (page 4; Figure S1). This is very interesting, but how about kernels for correct trials? A recent similar study showed that inter-trial-intervals affected decision strategies in mice (Iigaya, Fonseca, Murakami, Mainen, and Dayan, 2018). It would be interesting if the authors could discuss results for correct trials, too.

The reviewer brings up an important point: our original analysis was performed to make sure that the difference in behavior observed after correct and incorrect trials was not confounded by a difference in ITIs due to the after-error time-out. New results by Iigaya and colleagues suggest that ITIs is a variable of interest *per se* when studying the impact of history effects. We have performed additional analysis to address this point, which recapitulated in the new Supplementary Figure 2. Essentially, we found that indeed the length of the ITI had an impact on history biases but it did it only by modulating the lateral kernels *r+* and *r-*. Transition kernels remained unaffected. Together, this result emphasizes that these two biases are fundamentally different and the build-up-reset-and-rebound dynamics of the transition bias are a very robust feature of the behavior.

We have included a sentence in the Results (p. 5) presenting the result:

We also sought for other dependencies of *b* on the length of the inter-trial interval (ITI). After correct choices, *b* increased to more positive values for longer ITIs (Supplementary Fig. 2b left), but the sudden decrease to near-zero values after an error occurred for all ITIs (Supplementary Fig. 2b-c; see below).

We have then mentioned that the dependency on ITI is only via the lateral bias and not the transition bias:

Furthermore, only the lateral kernels showed a dependence on the ITI (Supplementary Fig. 2d).

Finally we have included a sentence in the Discussion with a reference to ligaya et al 2018:

Moreover, only the lateral bias showed a dependence on the length of the previous inter-trial interval (ITI) (ligaya et al. 2018): while both after-correct and after-error lateral biases became more positive for longer ITIs (i.e. favoring more repetitions), the transition bias remained unaffected (Supplementary Fig. 2d).

We also paste here the new Supplementary Fig. 2 with its caption:

Supplementary Figure 2. Repeating bias after different stimulus strengths and inter-trial interval (ITI) durations. **a**, Repeating bias b after error trials versus previous stimulus strength in the Alternating (red) and Repetitive blocks (blue). **b**, Repeating Bias b after correct (left) and error trials (right) versus inter-trial interval (ITI) in the Alternating (red) and Repetitive blocks (blue) for rats punished with a fixed 5 s after-error time-out (solid curve) or with a randomly interleaved 1, 3 or 5 s time-out (dotted curve). **c**, Proportion of Repeated responses (median across $n = 10$ animals) computed in trials following a correct (red and blue curves) or an error response (black curves). We computed each psychometric curve separated for short ITIs (left) and long ITIs (right) after previous choice, using a median split of the ITI of each animal after correct or error trial. The mean (25, 50, 75)% percentiles of the ITI after correct responses was (2.25, 2.68, 3.35) and after errors with 5 second time-out were (6.48, 6.89, 8.22). **d**, Average GLM weights (Group 1, $n = 10$ animals) fitted separately for trials after short ITI (bright colors) and long ITI (light colors). As in Fig. 4, responses were fitted separately for trials after-correct (orange colors) and after-error choices (black colors). The transition T^{++} kernel was the same for short and long ITIs, but the lateral kernels, r^+ and r^- , were more positive for longer ITIs. Thus, the transition bias build-up and reset is independent of ITI but lateral bias is more attractive after long ITI trials. In all panels dots show median across $n = 10$ animals (Group 1) and error bars the 1st and 3rd quartiles, except the control shown in panel c (dashed line) which corresponds $n = 3$ (subset of animals from Group 1).

1. 7. It is presumably good to have repeat/alternate biases in the authors' task. How do the animals' kernels compare with optimal bias kernels for the task, given their performance in perceptual decisions?

This is an interesting suggestion from the reviewer. We have now added a new Supplementary Figure in which we show what GLM kernels we would get if the responses were generated by the Dynamic Belief Model (Yu and Cohen 2008). This model assumes an optimal Bayesian observer which estimates the probability to repeat the previous stimulus from previous transitions.

We paste here the new Supplementary Fig. 16 and its caption:

Supplementary Figure 16. GLM for the optimal observer. We can formalize the optimal observer as building an expectation from previous trials of which side will be rewarded in the subsequent trial, and using this expectation as a prior to be combined with the stream of stimulus information (REF). Because such prior is updated on a trial-by-trial basis using the choices and outcomes from previous trials, it generates history-dependent choice biases that are captured by the transition and lateral kernels of the GLM. The precise form of those transition and lateral kernels will depend on how much knowledge of the structure of the task the observer has. A common assumption is that the observer does not know that the environment is structured in blocks of trials with a fixed repeating probability but she believes that there is a certain probability H (or *hazard rate*) that the probability to repeat the previous stimulus category changes to a new value randomly drawn from a fixed distribution (Yu and Cohen, 2008; Menzel et al, 2016). This is exactly the Dynamic Belief Model (DBM) studied in (Yu and Cohen, 2008) where they showed that the estimate of the repeating probability is very well approximated by a simple exponentially-weighted running average of recent repetitions and alternations of the *rewarded side*. In other words, in such a model the observer knows on every trial, independently of the response outcome, in

which side was the reward (i.e. the stimulus category) and can therefore track the probability that the rewarded side is repeated or alternated. The time constant of the exponential decay is $\tau = \log\left(\frac{3}{2(1-H)}\right)^{-1}$ (Yu and Cohen, 2008). **a-d**, Transition kernels obtained from the fitting of the GLM to synthetic data generated using the DBM. The exponential weighting of the DBM gives rise to exponential transition kernels that decay as a function of trial lag similarly to what we found for ++ transitions in our rats (Fig. 4b). However, there were multiple differences between the transition kernels displayed by the rats and the DBM. First, the DBM generated non-zero kernels in transitions involving error choices (i.e. T^+ , T^{+-} and T^-) simply because the optimal observer extracted the same information from all previous transitions independently of whether the choices were rewarded or not. Thus, the kernels for T^+ , T^{+-} generated by the DBM were the exact negative of the T^{++} kernel. This was because, for example a $R-R^+$ repetition ($T^+=1$) provided the same evidence favoring the alternation of the rewarded side than the $L+R^+$ alternation ($T^{++} = -1$). Using the same rationale, the kernel for T^- in the DBM was exactly equal to the T^{++} kernel. Second, the kernels of the DBM computed after error trials did not vanish (see black curves); instead, they were the negative mirror image of the corresponding kernels computed in after-correct trials (compare black and orange curves). This is simply because in the DBM accumulated evidence favored the repetition/alternation of the previous stimulus category c_{t-1} . In the GLM the transition evidence z^T , i.e. repetition $z^T > 0$ or alternation $z^T < 0$, was defined with respect to *the last response* r_{t-1} (i.e. the transition bias was defined as $\gamma^T = z^T \times r_{t-1}$). Because after errors the stimulus category and choice had opposite sign ($r_{t-1} = -c_{t-1}$), this caused the sign of the transition kernel after errors had to be reversed. For instance, if after a long series of correct repetitions there was an incorrect response, $R^+R^+R^+R^+R^-$ the DBM would be biased towards repeating the side of the last stimulus category L . This bias built from a series of repetitions, would be captured in the GLM as accumulated evidence to *alternate* ($z^T < 0$) the last response R . **e-f**, If the ideal observer was informed that the unconditioned probabilities to be rewarded in the Left vs the Right side were $P(L)=P(R)=0.5$, the DBM would display a null lateral kernel: the side of the rewarded response would not provide any information about the repeating probability. A more general ideal observer tracking both (1) the frequency rate of L vs R , (2) the repetition probability would generate a positive (negative) exponential kernel for the rewarded (unrewarded) responses (Wilder et al 2009, Mennyiel et al 2016).

We have also added a sentence referring to this figure in the Discussion section (p. 18):

Why does the transition bias reset after errors? The question is relevant because an ideal observer who was able to infer the location of the reward in correct and error trials would not show any reset (Yu and Cohen 2009; Meyniel et al 2016) (Supplementary Fig. 16).

Reviewer #2 (Remarks to the Author):

This is an interesting study examining how rats use priors and current sensory cues to make choices using behavioural measurements and computational models. Behavioural tasks are nicely designed and conducted, and the analyses are performed with care. The

fact that animals learn to repeat or alternate their choices depending on the block type is not surprising. However, the manuscript reports that immediately after error trials, the effect of priors goes away, and then comes back as soon as animals start to perform correctly. This aspect of the behaviour is interesting and novel.

2. 1. Given that the task has an active initiation for each trial, one question is whether animals take longer to initiate the trial after an error trial, compared to the other trials. It could be that after an error they take much longer to initiate the next trial, and they then show less effect of trial history. It is necessary to look into animals' initiation time to check for these and similar possibilities.

The reviewer brings up an important point. We think that the control the referee is proposing was addressed in our original submission by the current Supplementary Fig. 2b. There, we showed the magnitude of the repeating bias b after errors as a function of the duration of previous inter-trial interval (ITI). ITI was defined as the time interval between the outcome delivery in trial $t-1$ and the first nose poke in the center port in trial t . The figure shows that the reset effect after errors, $b \approx 0$, was maintained independently of the ITI.

Following the suggestion of Reviewer 1, we have now added a new analysis regarding the impact of the ITI on choice biases after-correct responses (Supplementary Fig. 2d). We have fitted separately the GLM analysis for short and long ITIs (median split) and found that the transition T^{++} kernel was the same in the two conditions.

2. 2. In addition to initiation time, choice reaction times should be analysed; Whether animals' choice reaction times show the same effect as choices? Could it be that after an error choices are reset but reaction times are not? It would be useful to investigate animals RT, in particular given the large number of trials collected and the fact that the task is a reaction time task.

We thank the reviewer for bringing up this point. We have now included the impact of the repeating bias in the rats' reaction time (RT) in a new paragraph in the Results section and two panels in Supplementary Fig. 1c-d. As it can be seen, reaction time was shorter for expected vs unexpected stimuli. Moreover, this modulation of the RTs vanished after errors further supporting the existence of an expectation reset (Supplementary Fig. 1c-d).

Further analysis of reaction times exceeds the scope of this article as we have found other unexpected effects on how Reaction Times are generated that can not be explained by a classical drift-diffusion models. The precise mechanisms underlying the generation of reaction times is actually the object of current in-depth study in our laboratory.

The new paragraph in the Results section is (p. 4):

“The expectation not only affected the rats' choice but also, modulated their reaction times (Supplementary Fig. 1c-d). After correct trials, the reaction time was shorter for expected stimuli (trials in where the repeating bias b was congruent with the block's tendency) compared to unexpected stimuli (trials in where the repeating bias b was incongruent with the block's tendency). Crucially, after error trials (black curves) the reaction time was not modulated by expectation. Hence, as for choices (ANOVA, *Block x Seq. Length x*

Prev.Outcome $F(6,250) = 3.06$, $p = 0.007$) the impact of repeating bias b on reaction time depended on previous trial outcome (ANOVA, *Block x Category(Rep.Stim.Evidence) x Prev.Outcome* $F(1,264) = 26.77$, $p = 1 < 1e-6$.)”

The RTs are shown in panels c-d of the Supplementary Fig. 1:

Supplementary Figure 1. Time course of Repeating Bias across trials and chronometric curves after correct and error trials. **a**, Time course of repeating bias b after correct (top) and error (bottom) responses through the course of one session starting with an Alternating block (left) or a Repetitive block (right). Repetitive blocks and Alternating blocks are indicated in blue and red color, respectively (vertical lines indicate block transitions). Curves show median b over $n = 9$ rats (one rat was excluded from the analysis because it only completed about 258 trials per session) computed using a 10 trials sliding window and 50 trials for after error trials. The shaded areas illustrate the 1st and 3rd quartiles. **b**, Fixed side bias B after error trials versus B after correct trials. Each dot represents one animal ($n = 10$) in Repetitive block (blue) or Alternating Block (red). **c-d**, Normalized Reaction time vs. repeating stimulus evidence during the repeating block (c) and the alternating block (d). Reaction time was separately computed after correct trials (red and blue curves) and after error trials (black curves). The reaction time was shorter for expected stimuli, i.e. trials in which the repeating stimulus evidence was congruent with the block's tendency, than for unexpected stimuli (*Block x Repeating Stimulus Category* $F(1,126) = 134.59$, $p < 1e-6$). In contrast, after error trials, when there was no defined expectation, reaction time was comparable for expected versus unexpected stimuli (black

curves; *Block x Repeating Stimulus Category* $F(1,126) = 0.02, p = 0.88$). Significance was assessed using a mixed-effects ANOVA using as factors Block-type (Repetitive / Alternating Block), repeating stimulus category (evidence to repeat / alternate previous response), stimulus strengths and random factor animal. For the joint analysis of the after-correct and after error reaction time, previous trial outcome (correct/error) was also included. The curves show mean normalized Reaction Time across $n = 10$ animals (Group 1). To mitigate the variability of reaction times across animals, reaction time was normalized by dividing, separately for each animal, all times by the maximum average reaction time among all conditions. Error bars show the standard error.

2. 3. In this type of experiments, error trials could happen more frequently towards the end of the session, as the motivation slowly drops. If this is the case in these experiments, then it would be necessary to perform analysis which takes this into account and/or repeat plots such as figure 2 only for late trials of the sessions.

We thank the reviewer for her/his concern about possible confounds in our analysis of post-correct vs post-error trials due to possible fluctuations in the proportion of correct vs error trials within sessions. Actually, there was no significant difference between average accuracy in the last 200 trials of each session vs average accuracy in other trials (average 75.12% for last 200 trials vs 74.68% for the rest of trials; two-tailed paired t-test across animals, $p = 0.30$). The effect was not significant either when focussing on the last 100 or 150 trials in each session. In any case, we have included in Supplementary Fig.1a the repeating bias after error trials showing that the reset effect is independent of the location of the error in the experimental session.

2. 4. Similarly, it is reasonable to think that several of errors are actually happening right after the block switch. Thus, it would be useful to compare error trials and correct trials that happen at the onset of the blocks.

The reviewer is right: rats performed slightly worse after block switch: average accuracy in the first 50 trials of a block (73.05%) was lower than during the rest of the block (75.51%; paired t-test: $p < 10^{-5}$). This can be appreciated in the figure below.

Although the difference in accuracy is relatively small, we addressed the reviewer's concern by running the GLM model only for the subset of trials at the onset of blocks (first 50 trials in a block). We added a new Supplementary Fig. 10 that shows that the pattern of GLM weights was very similar at the block onset compared to weights obtained from all trials. This shows that the observed dissociation in behavior following correct and error trials is not confounded by variation of the proportion of correct and error trials within blocks.

We have added the following sentence in the Results addressing this control (p. 10):

“Importantly, the weights were identical when computed separately in repetition and alternation blocks (Supplementary Fig. 9) or for the trials at the beginning of the blocks when the accuracy was smaller (accuracy of trial ranges 1-50 and 50-200 were 0.73 and 0.76, two-tailed paired *t*-test $p < 1e-5$; Supplementary Fig. 10).”

Here is the new Supplementary Fig. 10 showing the comparison of GLM weights between block onset and all trials:

Supplementary Figure 10. Rats use the same strategy at the beginning and at the end of the blocks. Panels are as in Supplementary Fig. 9 but comparing the GLM weights fitted to all trials (light colored curves) and to only the first 50 trials of each block (bright colored curves). Accuracy at the beginning of the block was lower than during the rest of the block (trials 1-50 yielded an average accuracy of 0.730 and trials 50-200 yielded 0.755; two-tailed paired *t*-test $p < 1e-5$). This difference was probably due to animals not having yet accumulated sufficient transition evidence congruent with the corresponding block (e.g. in switching from a Repetition block to an Alternation block they had to change z^T from positive to negative). And yet, the strategy they used during the block transition was the same as revealed by the similar GLM weights. Accuracy did not change significantly at the end of the session (average 0.751 for last 200 trials vs 0.747 for the rest of trials; two-tailed paired *t*-test across animals, $p = 0.30$; similar results were obtained when focussing on the last 150 or 100 trials of each session). Points in all panels show median coefficients across animals and error bars indicate first and third quartiles.

2.5. Lastly, I am wondering whether it would make sense to move the first regression section (Figure 4) to the Supplements. I am suggesting this because I found it hard to

digest that regression and then follow the main model. This is just a suggestion as authors might find it necessary to have both analyses in the main text.

The referee is right that the GLM model (Fig. 4) might be hard to digest for some readers. However, we prefer to maintain this analysis because provides very strong evidence of the existence of the reset of the transition bias. Although perhaps in the community of Reinforcement Learning this type of GLM is not so common, we feel that it has come a standard description of the factors impacting choices in 2AFC tasks with a minimal number of assumptions. It has been widely used in the subfield of decision making both in free choice tasks (see e.g. Corrado et al., 2005; Lau and Glimcher, 2005) and perceptual decision tasks (Abrahamyan et al., 2016; Braun et al., 2018; Busse et al., 2011; Fründ et al., 2014; Nogueira et al., 2017; Urai et al., 2017).

We think that part of the problem with the GLM was because the transition between the conceptual model in Fig. 3 and the GLM model in Fig. 4 was not very smooth (as pointed by Reviewer 1). We have now modified Fig. 3 and the text presenting Fig. 4 so that the link between these two figures is easier and the readers can understand the GLM with less effort. We have also changed the Supplementary Fig. 5 with the description of the GLM to make it more clear (sensory module has been separates, kernels are now explicitly defined, we introduced the transition *evidence* z^T , ...).

Also note that Reviewer 3 is actually suggesting the contrary: to expand the explanation of the GLM by bringing the (now) Supplementary Fig. 5 (detailed description of the different regressors) as a main figure, something we will refrain from doing given the this comment (i.e. comment 2.5).

Minor:

2. 6. Figure 5C benefits from some legends describing the colours.

We agree with the reviewer. We have added the legend to the figure.

2. 7. Figures 6b-f also benefit from such legends.

We agree with the reviewer. We have added the legend to the figure and indicate added choice type (with the color code) of each bar in each subplot.

Reviewer #3 (Remarks to the Author):

In this manuscript, Hermoso-Mendizabal and colleagues investigate the impact of past-trial influences on auditory perceptual decisions, using a combination of behavioral experiments and various modelling approaches. Investigating psychometric curves, they first demonstrate that psychometric curves are influenced by the repeat/transition probabilities of the current block of trials, but only after a correct previous trial. To get insights into the

specific factors underlying this effect, they then use a GLM, with which they can dissociate impact of previous responses (lateral bias) vs. previous repetitions/alternations (transition bias). Analysis of the fitted coefficients reveals that the effect is driven by previous transitions, but again only if responses were correct. Third, investigating the correlation of transition bias across lags (transfer coefficient), the authors can demonstrate that the transition bias is reset after an error, but re-emerges following a correct response afterwards. These insights are further corroborated by a generative model, which could capture the effects of past stimuli and outcomes on internal variables, but also successfully predicted the animals' biases in simulations.

Overall, the manuscript reveals fundamental insights into overarching principles underlying behavior in a 2AFC task, which will be of general interest to circuits neuroscientists working in animal models, cognitive neuroscientists working in humans, and those interested in behavior.

MAJOR

3. 1. Supplementary Fig. 4: I would strongly recommend moving this figure to the main text. The GLM is a very powerfully laid out to not only capture psychophysical kernels, past-trial sensory adaptation effects, but also strategies like “win-stay”, “loose-switch”. Hiding all of this in the supplementary material not only makes the text hard to follow, but would also be a pity. While the figure is very helpful, several things regarding the GLM remain unclear, also because the text in supplementary material is a bit more raw than for the main text.

We thank the reviewer for this advice and his/her positive words about the power of the GLM. Whether to include the GLM figure into the main text has been a matter of debate between us since the first drafts of the paper. Although we concur with the reviewer that understanding the details of the GLM can be very clarifying, we have also noted that for some readers this takes a lot of effort and they are “exhausted” by they arrive to the latent variable model. Comments from Reviewers 1 (see 1.5 above) and from Reviewer 2 (see 2.5 above) suggest somehow the opposite: to remove the GLM altogether. For these reasons we prefer not to bring the Supplementary Fig. describing the GLM to the main text. We have done however a few changes that might bring ease the understanding of the details of the GLM:

- We have modified Fig. 3 showing the conceptual separation of lateral and transition biases to facilitate the link to the GLM (the figure now shows in panel **b** a weighed sum of previous responses and transitions). Here is the new Fig. 3 with its caption:

Figure 3. Dissecting two different history choice biases. Cartoon of an example series of four choices, $R^+R^+R^+L^+$, illustrating the build-up of the lateral and transition biases. **a**, The lateral bias, capturing the tendency to make Rightward or Leftward responses, increases towards the Right in the first three R^+ trials and compensates this build-up with the last L^+ response. Its net impact on the final trial is a Rightward bias. **b**, Schematic of the sequence of rewarded rat responses showing the transitions, defined as the relation between two consecutive responses, being Repetitions (Rep, blue arrows) or Alternations (Alt, red arrow). The animal computes each choice from combining its expectation based on a weighted sum of previous alternations (bottom gray balloon) and previous responses (upper gray balloon) with the current stimulus sensory information (see last trial). **c**, Transition evidence z^T captures the tendency to repeat or alternate the previous response based on the series of previous transitions Rep+Rep+Alt predicting a Repetition in the final trial. **d**, The transition bias γ^T is obtained by projecting the Transition evidence z^T (c) onto the Right-Left choice space via a multiplication with the previous response r_{t-1} (see gray arrow). **e**, The evidence provided by the current stimulus is summed to the addition of the biases $\gamma^L + \gamma^T$ and passed through a sigmoid function, yielding the probability of selecting a Rightward response (Supplementary Fig. 5).

- We have change the text describing of Fig. 4 with the GLM results so that it is easier to understand for readers which are not familiarized with GLMs (Results section, p. 8):

“To quantitatively assess how subjects computed these biases we used an explorative approach that assumed that, in each trial, animals combined linearly the responses r ($r = R, L$) and transitions T ($T = Rep, Alt$) from the last ten trials in order to generate the lateral and transition bias, respectively (see Fig. gray boxes in Fig. 4c and Supplementary Fig. 5; see Supplementary Methods section 2 for details). The two biases were then combined with the stimulus evidence in order to yield a decision (Fig. 4e). Fitting such a generalized linear model (GLM) to the behavior of a rat implied finding the weight with which each of these past (e.g. previous transitions) and current events influenced the animal choices

(Abrahamyan et al., 2016; Braun et al., 2018; Busse et al., 2011; Fründ et al., 2014; Nogueira et al., 2017; Urai et al., 2017). “

- We have also changed the Supplementary Fig. 5 with the description of the GLM to make it more clear (sensory module has been separated, kernels are now explicitly defined, we introduced the transition $evidence z^T$, ...).

3. 2. * - Supplementary Material, p. 4 “The net stimulus evidence of each of the previous trials ($t-1$, $t-2$, ..., green and purple bars in gray box) are weighted by the previous stimulus kernel”. What exactly is “previous stimulus kernel”? In the text, it says: sums of stimulus evidence over all frames, which sounds more straightforward.

We thank the reviewer for picking this. We have now changed the name of the “Previous Stimulus kernel” to “After-effect” kernel everywhere in the paper. The quoted sentence now reads (caption of Supplementary Fig. 5, p. 6 Supp. Materials):

c, The net stimulus evidence, i.e. the sum of the stimulus evidence over all frames, of each of the previous trials ($t-1$, $t-2$, ..., green and purple bars in gray box) is weighted by the after-effect kernel providing the After-effect bias γ^A_t .

3. 3. * - Supplementary Material, p. 14: description of Lateral module: what is i , j ?

Sorry, this was not explicit: it was just in the definition of the delta Kronecker, which takes any two variables i and j as input and outputs 1 if $i=j$ and 0 otherwise. This has now been corrected.

3. 4. Can the authors please explain in more detail why they chose to group regressors into specific sums representing the various modules?

The modules were introduced for illustration purposes as all the regressors are actually fed directly into the GLM (there are no intermediate sums computed). We have included the following sentence in the caption of Supplementary Figure 5 (p. 6 Supp. Material):

Because all regressors are ultimately summed together, the modules have no functional relevance and were only introduced for structural clarity.

Moreover, we have splitted the “Sensory Module” into the “Current Stimulus module” and the “After-effect module” (see Suppl. Fig. 5). We think this separation is also going to make more clear that the modules are just grouping regressors of the same type.

3. 5. Can the authors please also clarify the use of variance to estimate each module’s estimated bias (the beginning of sections 2.2 would be informative even earlier in the text, i.e. after the description of the transition module).

We have added the following explanation in the text:

“ The impact of effective stimulus evidence γ^S_t (Eq. 1) and the history biases γ^A_t, γ^L_t and γ^T_t (Eqs. 2-4) directly depends on how much each term varies from trial to trial, i.e. how much

the corresponding variable γ_t^X ($X=S,A,L$ and T) will nudge to towards a right choice bias on some trials, a left choice bias on other trials. “

Regarding the organization of the material, we thank the reviewer for his/her suggestion. We have made major changes in the way we now describe the different terms of the GLM following his/her suggestion. In particular, we introduce the kernels and the “bias variables” γ_t^X much earlier in the description (see page 19 - 21 in the Supplementary Material). We think these changes will facilitate the understanding for most readers.

3. 6. Similar issues apply to Fig. 5 and associated supplementary text: Please move some of the explanation of what a transfer coefficient is from figure legend to proper text, and streamline the associated supplementary text to help the reader find the appropriate information. Section 2.5 of the supplementary material almost starts like a second version of the main text, which is not helpful. The explanation of transfer coefficients (in supplementary material) is not easy to follow and seems still raw.

We have now defined the Transfer Coefficient in the main text and referred to the corresponding section of the Supp. Methods for details. This is the new sentences in the Results section (p. 11):

“To test these two hypotheses, we defined the transfer coefficient $\gamma_t^T \rightarrow \gamma_{t+k}^T$ that quantified how, given the value of the bias in trial t , γ_t^T , we could predict the bias in trial $t+k$, γ_{t+k}^T (see section 2.6 in Supplementary Methods for details). The transfer coefficient was computed as a function of trial lag k conditioning on the sequence of outcomes (Fig. 5c).”

We have also completely rewritten Section 2.5. in the Supplementary text (it is now Section 2.6.). The new version is (i) more connected with the flow of the main text, (ii) defines the transfer coefficients much earlier in the section, (iii) the broad rationale of the analysis is more clearly explained at the beginning of the section and (iv) the steps taken to fit this meta-GLM more orderly described.

3. 7. Maybe I missed it, but in fitting the GLM or other models, was there some cross-validation involved?

The reviewer is raising an important question. Our original take on this point was that, due to the very high number of trials per animal (average ~50.000), the regularization hyperparameters would have only a marginal impact on estimated parameters. Because our fitting algorithms are computationally expensive, we did not apply cross-validation to select this parameter for time and environmental considerations. We chose to set the ridge hyperparameter $\lambda=1$ (in the latent variable model, this hyperparameter was termed σ). This corresponds to a gaussian prior with standard deviation of 1 for each weight, consistent with the scale of weights (in the 0.01-2 range). We have now run a series of control analyses to address the reviewer’s concern. First, we have fitted both the GLM and the dynamical latent model with different values of the ridge hyperparameter: $\lambda = 0, 0.1, 1, 10$ and 100. We could indeed confirm that, except for the highest value (100), the estimated model parameters varied very little depending on the value chosen for λ (see plot below). For the GLM (Figure 3.7a below), the correlation between parameters fitted with $\lambda=1$ and $\lambda = 0$ was $\text{Corr}(\lambda=1, \lambda = 0)=0.9998$. Moreover, $\text{Corr}(1, 0.1)=0.9998$ and $\text{Corr}(1, 10)=0.997$. For the dynamical model (Figure 3.7 b below), $\text{Corr}(1, 0)=0.9480$, $\text{Corr}(1, 0.1)=1$ and

Corr(1, 10)=0.985. Second, to estimate the goodness of fit as a function λ (for $0 < \lambda < 100$), we computed the approximated model evidence corresponding to each value of λ , and for each animal (using the Laplace approximation). This method is less computationally intensive than cross-validation and provides similar results for high number of observations. The best fit was obtained for $\lambda=10$ for all rats in the GLM (see Fig. 3.7 c). For the latent model, the fits with $\lambda=1$ and $\lambda=10$ were similarly good (Fig. 3.7d). Since a value of $\lambda=10$ provides very similar results to a value of $\lambda=1$ in both models, we conclude that the value of $\lambda=1$ was a reasonable choice in the first place.

Figure 3.7

To acknowledge this line of arguments, we have now added in the Supplementary Methods section 2.1 (p. 22):

“Because we had a very large number of trials per animal, the exact value of the hyperparameter had very limited influence on the estimated weights, provided this value was not too large ($\lambda < 100$). When λ was ≥ 100 the goodness of fit decayed dramatically as could be assessed by comparing approximated marginal evidence using the Laplace approximation [Bishop, 2006].”

And a similar sentence has been added in section 3.1 of the Supp. Methods regarding the latent variable model (p. 26).

“Because of the very large number of trials per animal, the exact value of sigma had little influence on the estimated weights up to values sigma < 100 . When sigma was ≥ 100 the goodness of fit, assessed by comparing approximated marginal evidence [Bishop, 2006], decreased dramatically.”

3. 8. It might be interesting to compare simulations by the GLM and the generative model to get insights into how performance of these two approaches compare.

The original Supplementary Figure 13b (now Supp. Fig. 14b) provided the model comparison between the GLM and generative approach. We found that for all but one rat, the generative model provides a better account of behavioral data, probably because it can capture most of the influence of stimuli and recent history on rat choices with much fewer parameters. However, following the reviewer’s advice, we sought to characterize more qualitatively how well the dynamical model could explain the form of the different kernels found in the GLM. To do so, we simulated the dynamical model for each rat (as performed for Figure 7) and then applied the GLM analysis on this simulated data. The results are now presented in the new Supplementary Fig. 16 (shown below). The figure shows that the GLM fitted on this simulated data captured most of the properties found on experimental data: the shape of T++ weights after correct response; the T++ weights turning null if previous trial was an error; the very small weight for other types of transition (T+-, T-+ and T—); the win-stay-low-switch strategy exposed by the lateral weights (L+ and L-). These results provide further evidence that our dynamical model captured with few parameters the essence of rat behavior. We have included the figure as Supplementary Fig. 15 and added the following reference in Results section (p. 14):

“Finally, subsequent fitting of our GLM to the simulated data showed that the dynamical model was able to reproduce the different history kernels displayed by the animals (Supplementary Fig. 15).”

This is the new Supp. Fig. 15 showing the GLM kernels of the simulated data:

Supplementary Figure 15. GLM analysis fitted to simulated data from standard dynamical model. For each rat, we fitted the same GLM used with the experimental data (Fig. 4) to the simulated data from the standard dynamical model. Panels show the different GLM kernels as described in Figure 4.

MINOR:

3. 9. p. 3: “repeating psychometric curves; sensory evidence in favour of the repeating choice”: this is impossible to understand based on results alone, and only the example in

the methods section really helps. I would encourage the authors to re-phrase and/or expand the explanation in the results session

We concur with the referee that this sentence was too obscure. We have now expanded the explanation of this important Repeating Psychometric Curve in the Results (p. 3):

“To estimate the impact on choice of the serial correlations of the stimulus sequence, we also analyzed the *repeating psychometric curves*, showing the proportion of trials where the animals repeated the previous choice as a function of the sensory evidence in favor of the repeating choice (e.g. a Rightward stimulus with evidence e after a Left response implied a repeating stimulus evidence equal to $-e$). The horizontal shift of this new psychometric curve, parametrized by the repeating bias b , measured the history-dependent tendency to repeat or alternate the previous choice.”

We have also clarified the distinction between the standard Psychometric Curve quantifying the proportion of Rightward responses from the novel Psychometric curve quantifying the proportion of *Repeated* responses in the Methods section (p. 20-21):

“We calculated (1) the proportion of Rightward responses vs. stimulus evidence e (Fig. 1a left) and (2) the Proportion of Repeated responses as a function of Repeating Stimulus Evidence \hat{e} defined in the t -th trial as $\hat{e}_t = r_{t-1}e_t$, with $r_{t-1} = \{-1, 1\}$ representing whether the response in the previous trial was Left or Right, respectively (Fig. 1b). Thus, positive or negative Repeating Stimulus Evidence denote trials in which the animals had evidence to repeat their previous choice. In other words, a Rightward stimulus with evidence $e = +0.23$ after a Left response implied a repeating stimulus evidence equal to -0.23 . Both psychometric curves were separately fitted to a 2-parameter probit function (using Matlab function *nlinfit*):

$$P_{\text{Rightwards}}(e) = \frac{1}{2} \left(1 + \operatorname{erf}\left(\frac{\beta e + b}{\sqrt{2}}\right) \right) \quad (2)$$

$$P_{\text{Repeat}}(\hat{e}) = \frac{1}{2} \left(1 + \operatorname{erf}\left(\frac{\beta \hat{e} + B}{\sqrt{2}}\right) \right) \quad (3)$$

The sensitivities β and β' quantified the stimulus discrimination ability, while the Fixed side bias B captured the animal side preference for the Left ($B < 0$) or Right port ($B > 0$), and the repeating bias b captured the animal's tendency to repeat ($b > 0$) or alternate ($b < 0$) their previous choice. “

3.10. 2c: I would encourage the authors to plot the repeating / fixed side bias on the same scale since it will allow better judging of their relative sizes

We thank the reviewer for the suggestion; we have updated the axis scale.

3.11. 2e: is the difference between the repeating bias b for repetitions vs. alternations significant? Please modify statement “This plateau was greater after repetitive patterns rather than alternating patterns” to include the statistics. Also, the onset seems different. Can the authors please comment?

The absolute value of the plateau is 0.96 for the repetitive sequence and 0.77 for alternating sequence, but the difference was not significant ($p = 0.0953$; mean last three points and paired t-test). Because of that, we have removed this comparison of the plateaus from the text.

Regarding the block asymmetry of b at the “onset” of these curves (i.e. compare red and blue points for sequence length $n = 2$ in Fig. 2e), it is now explained around Figure 6, where the same plots of repeating bias b vs sequence length n , are dissected into the lateral and transition components. We have added the following sentence to explain this difference in the Results (p. 14):

“In particular, the block asymmetry in b after the first correct choice (i.e. after an EX+ sequence) could be explained by the two contributions having the same or opposite sign (see $n=2$ in Fig. 7a) ...”

3.13. p.4: increased with n (n what? I assume trials)

This second part of the sentence “and it increased with n ” was a typo and has been removed.

3.14. p. 4: “The impact on performance was largest for low stimulus strength etc.” Can the authors please provide the statistics and explain explicitly how they came to the conclusion?

We thank the reviewer for the precisions. We realized that the sentences referring to the impact of the repetition bias on accuracy were not very clear and lacked the proper statistical tests. We have decided to rewrite the two original sentences touching on the effect on accuracy which have now become a new small paragraph on the Results (p. 5):

“The trial-to-trial updating of the response prior had a direct impact on the animals categorization accuracy. Overall, the repeating bias seemed advantageous for the task, as the averaged categorization accuracy was higher for trials following a correct trial than for trials following an error, in which b was reset to zero (0.76 versus 0.72, $p < 1e-04$ two-tailed t -test). However, the repeating bias increased the subjects accuracy when it was congruent with the block tendency but it decreased accuracy when it was incongruent with it (Supplementary Fig.4a-d). Moreover, in the congruent condition, the impact of the prior on accuracy was largest for low stimulus strength and it vanished to zero as the stimulus strength increased; this interaction was however absent in the incongruent condition where the detrimental effect of the prior was uniform across all stimulus strengths (see Supplementary Fig.4a-d for details). “

We also made modifications in the Supplementary Fig.4 where we include all the ANOVAs performed to measure the effects of repeating bias on accuracy :

Supplementary Figure 4. The repeating bias build-up from previous trials impacts the rats' classification accuracy. Accuracy, defined as the fraction of correct responses, versus the current stimulus strength s in trials following different trial sequences. **a-b**, Congruent condition in which accuracy was computed in the Repeating Block for trials following several correct repetitions (**a**) or in the Alternating block following correct alternations (**b**). Insets show the color code for sequences of correct repetitions (red) and alternations (blue). Accuracy increased with the length of the congruent sequence, particularly at low stimulus strength (see statistics below). **c-d**, Incongruent condition in which accuracy was computed in the Repeating block following correct alternations (**c**) and in the Alternating block following correct repetitions (**d**). In contrast to the congruent condition, the detrimental effect on accuracy of increasing the length of the incongruent sequence was independent of stimulus strength. Significance was assessed using a mixed-effects ANOVA with factors stimulus strength s , sequence length n , prior-block congruency (congruent/incongruent) and random factor animal, revealed a significant three-way interaction $n \times s \times$ congruency, $F(1,786)=38.57$, $p < 1e-9$; separate analysis of the congruent and incongruent conditions yielded a significant two-way interaction $n \times s$ for the congruent condition ($F(1,387)=42.09$, $p < 1e-9$) but not for the incongruent condition ($F(1,387)=2.12$, $p=0.146$). Dashed lines mark the maximum and minimum performance that can be reached for stimuli with $s = 0$ in each block if subjects adopted the strategy of always repeating or alternating the previous response. In the repeating block these values are the sequence repeating probability P_{rep} and $1 - P_{rep}$, respectively, whereas in the alternating block they are $1 - P_{rep}$ and P_{rep} . Dots show median across $n = 10$ animals (Group 1). Error bars and shaded areas show the 1st and 3rd quartiles.

3. 15. p. 7: "This is achieved by multiplying zT with the last response $rt - 1$," - I would recommend adding here that this does not necessarily have to apply, but happens in the authors' model, i.e. "This is achieved in our framework..." or something along these lines

p. 7: “Lateral and transition biases have an opposite influence in the final choice” - I would recommend adding “In this example, lateral ...” - they can also have congruent influences, e.g. in the decisions before the final choice

We thank the reviewer for the precisions. We have accepted both suggested changes.

3. 16. p. 7: “... since it allows to capitalize the sequence correlations in both types of blocks” - consider rephrasing, this is not entirely clear (what exactly is ‘it’? what does ‘capitalize’ mean in this context?)

We thank the reviewer the observation; we have rewritten the sentence:

“Although only the transition bias is adaptive in the task, since such bias allows to take advantage of correlations between successive reward sides in both types of blocks, the two biases could in principle contribute to the repeating bias b described above.”

3. 17. p. 7: see Methods for details -> Supplementary Methods - it took me a while to find...

This has been changed to “see Supplementary Methods section 2 for details”, thank you for the correction.

3. 18. p. 8, Figure legend: “Influence of the response side (Left vs Right) from previously rewarded (r^+ , left panel) and unrewarded (r^- , right panel) trials”. Shouldn’t this say “Influence ON the response side (Prob Right) OF previously ...”?

Formulation was not clear, we have changed to :

“GLM weights of previously rewarded (r^+ , left panel) and unrewarded (r^- , right panel) responses. These kernels quantify the influence on choice of the side (Left vs Right) of previous responses.”

3. 19. p. 7: “The contribution of each response to γ^L depended on its outcome following a win-stay-lose-switch pattern” - please rephrase, this is unclear.

We agree it was not clear, and have reformulated to:

“The contribution to γ^L of each response depended on whether the response was rewarded or not, following a win-stay-lose-switch pattern”

3. 20. Fig. 4a: Maybe I missed this in the main text, but please explain with a quick sentence and reference to supplementary material (Indeterminacy analysis) why there are data missing from certain trial lags.

We thank the reviewer for the suggestion. We have added to the legend:

“Some weights at trial lags 1 and 2 are not shown because of existing indeterminacies between regressors (see Supplementary Methods Section 2.3).”

3. 21. Fig. S5 legend: “This repulsive side bias captured the effect of having heard the previous exact stimuli, above and beyond the bias introduced by the category of those stimuli which was captured by the lateral bias” -> previous exact stimuli is unclear; above AND beyond

- p. 15: “go beyond from previous reports” -> go beyond previous

- p. 18: “(see below)” misplaced here

- Supplementary Material, p. 14: “variability on” -> variability of

All the sentences have been corrected. Thank you.

3. 22. Supplementary Material: “Compared with the GLM that required that rats maintained memory of all the features for each of the 10 previous trials in order to make a decision, this latent variable model reduced the working memory load to simply maintaining the value of three latent variables.” - This seems odd - the GLM has no influence whatsoever on the rats’ behavior.

We agree on the ill formulation. This was changed to :

“Compared with the GLM, an implementation of which would require the subjects to maintain in memory all the features for each of the 10 previous trials in order to make a decision, implementing the latent variable model would reduce the working memory load to simply maintaining the value of three latent variables.”

Thank you very much for such a careful reading of the manuscript and about so many suggestions that have substantially improved it.

Reviewers' Comments:

Reviewer #1:

Remarks to the Author:

The authors fully addressed my concerns.

Reviewer #2:

Remarks to the Author:

The authors have addressed all my questions, and the paper has been substantially improved.

Reviewer #3:

Remarks to the Author:

The authors have addressed all my previous comments.

To further improve the clarity of presentation, I have a few more minor comments:

- l. 128: "animals side preference" -> animals' side preference
- Fig. S1: "computed using a 10 trials sliding window and 50 trials for after error trials": The analysis looks very convincing, but I can't follow how exactly it was conducted. Why is the sliding window size different for correct vs. error trials? Why are the transitions different between top and bottom - is trial number maybe 'correct trial number' and 'error trial number'?
- l. 139-147: I find the statistical reporting chosen here confusing, as the statistics for half of the results can only be found in the supplemental information, and the interaction in l. 146 is mentioned where one would expect the post-hoc comparison. For clarity, I would recommend stating the statistics also in the main text, and maybe removing the single reference to 'black curves' in the main text.
- l. 164: in a trial-by-trial basis -> on a trial-by-trial basis
- l. 214: see gray boxes in Fig. 4c: not sure what this refers to
- l. 216: Fig. 4e -> Fig. 3e?
- l. 244: "that rats behavior" -> rats' behaviour
- l. 247: "the magnitude of transition kernel was much more homogenous across animals than the lateral kernel (Supplementary Fig. 7)." - please report an appropriate statistical test
- Fig. S6, legend: "on the rats choices" -> rats' choices
- l. 274: "rats decisions" -> rats' decisions

We here address below the comments of each referee (rewritten **in black**), providing explanations and clarifications (**in blue**) and pasting the new additions and edits to the paper that address these comments (colored **in red** in this document and the main text).

Reviewer #1:

The authors fully addressed my concerns.

We thank the reviewer for all his/her previous comments.

Reviewer #2:

The authors have addressed all my questions, and the paper has been substantially improved.

We thank the reviewer for all his/her previous comments.

Reviewer #3:

The authors have addressed all my previous comments.

We thank the reviewer for all his/her previous comments.

To further improve the clarity of presentation, I have a few more minor comments:

- l. 128: "animals side preference" -> animals' side preference

Corrected. Thanks.

- Fig. S1: "computed using a 10 trials sliding window and 50 trials for after error trials": The analysis looks very convincing, but I can't follow how exactly it was conducted. Why is the sliding window size different for correct vs. error trials? Why are the transitions different between top and bottom - is trial number maybe 'correct trial number' and 'error trial number'?

Thanks for mentioning this.

We used two different sliding windows because for errors there are much fewer trials (i.e. ratio one to four). The curve after-error using sliding window 10 trials looks just the same but significantly noisier. We have added the following clarification in the caption of the figure explaining the choice of different sliding windows:

Curves show median b over $n = 9$ rats (one rat was excluded from the analysis because it only completed about 258 trials per session) computed using a 10 trials sliding window and 50 trials for after error trials (simply because there were much fewer error trials).

Regarding the alignment of the two plots, this was a typo. Because the sliding windows were different the first and last point in each plot corresponded to a different (centered) trial: e.g. the first point was trial 10 for after-correct and trial 50 for after-error. This has been corrected and now the trial axes in both plots coincide.

- l. 139-147: I find the statistical reporting chosen here confusing, as the statistics for half of the results can only be found in the supplemental information, and the interaction in l. 146

is mentioned where one would expect the post-hoc comparison. For clarity, I would recommend stating the statistics also in the main text, and maybe removing the single reference to 'black curves' in the main text.

The referee is right about this paragraph being a bit too short to understand the statistics perform to support the reports about Reaction Times. We have now brought the ANOVA results from the Supp Figure S1 to the main text. Also, we have performed the required post hoc analysis on Expected vs Unexpected RTs in the after-correct condition (the after-error condition does not require a post hoc because the interaction between block (Repeating/Alternating) and Repeating Stimulus Category (Repeating/Alternating stim. category) is not significant (reported). The paragraph including the required stats reads now like this:

The expectation did not only affect rats' choices but also modulated their reaction times (Supplementary Fig. 1c-d). After correct trials, the reaction time was shorter for expected stimuli (i.e. trials in which the repeating stimulus evidence was congruent with the block's tendency) compared to unexpected stimuli (i.e. trials in which the repeating stimulus evidence was incongruent with the block's tendency; ANOVA *block x Repeating Stimulus Category* $F(1,126) = 134.59$, $p < 1e-6$; mean normalized $RT(\text{expected}) - RT(\text{unexpected}) = -0.10$, post-hoc two-tailed paired t -test $p < 1e-10$). Crucially, after-error trials the reaction time was not modulated by expectation (ANOVA, *block x Repeating stimulus category x previous outcome* $F(1,264) = 26.77$, $p < 1e-6$; separate analysis for the after error condition yielded *block x Repeating Stimulus Category* $F(1,126) = 0.02$, $p = 0.88$). Hence, as for choices, the impact of repeating bias b on reaction time depended on previous trial outcome, being absent after error trials.

- I. 164: in a trial-by-trial basis -> on a trial-by-trial basis
Corrected. Thanks.

- I. 214: see gray boxes in Fig. 4c: not sure what this refers to
This was a typo. We meant: "see gray boxes in Fig. 3b".
It has now been corrected. Thanks.

- I. 216: Fig. 4e -> Fig. 3e?
Corrected. Thanks.

- I. 244: "that rats behavior" -> rats' behaviour
Corrected. Thanks.

- I. 247: "the magnitude of transition kernel was much more homogenous across animals than the lateral kernel (Supplementary Fig. 7). " - please report an appropriate statistical test.

We performed the corresponding statistical test and now report it in the main text and in the Supp. Fig. 7. In the main text we have included the following:

In fact the magnitude of transition kernel was much more homogenous across animals than the lateral kernel (after-correct T^{++} vs. r^+ , $F(9,9) = 0.24$, $p < 0.05$; after-correct T^{++} vs. r^- , $F(9,9) = 0.094$, $p < 0.002$; see Supplementary Fig. 7).

In the caption of the Supp. Figure 7 we have included how we calculate this comparison:

We tested whether the heterogeneity across animals of the T^{++} kernel after-correct responses was smaller to the heterogeneity of the r^+ or the r^- kernels after-correct responses. For that, we normalized the individual T^{++} kernels by the maximum value across lags and animals and then we summed all the time lags in each animal. We did the same for the r^+ and r^- kernels. The variance of the summed normalized T^{++} kernels was significantly smaller than of the r^+ kernels (variance ratio was 0.24, $p = 0.0425$, two-tailed F -test) and also when comparing between T^{++} and r^- (variance ration was 0.094, $p = 0.0016$, two-tailed F -test). We did not compare the T^{++} individual kernels after errors because their magnitude was negligible.

- Fig. S6, legend: "on the rats choices" -> rats' choices

Corrected. Thanks.

- l. 274: "rats decisions" -> rats' decisions

Corrected. Thanks.